# Bridging the Stability-Expressivity Gap: Synthetic Data Scaling and Preference Alignment for Low-Resource Spoken Language Models

Yizhong Geng [1]  Yanliang Li [2]  Jinghan Yang [1]  Tianhan Jiang [3]  Boxun An [4]  Ya Li [1]  Xiaoyu Shen [5]

## Abstract

Spoken Language Models (SLMs) have emerged as a promising paradigm for speech synthesis by bypassing explicit grapheme-to-phoneme pipelines. However, their effectiveness in low-resource languages remains fundamentally limited by the scarcity of transcribed speech. In practice, synthetic data has become the primary strategy for scaling SLMs in such settings, providing reliable phonetic supervision when real data is insufficient. In this work, we show that this reliance introduces a fundamental trade-off, which we term the Stability-Expressivity Gap: while synthetic data improves phonetic accuracy, it progressively suppresses prosodic variability, ultimately leading to a collapse of expressivity (Synthetic Erosion). To bridge this gap, we propose two self-alignment frameworks. Disentanglement-Guided Self-Alignment (DGSA) recovers expressivity for complex languages by exploiting prosody-timbre separation. For regimes where authentic references are exceptionally limited, Temperature-Driven Self-Critique (TDSC) stabilizes generation through automated exploration and filtering. Our approach outperforms strong commercial systems, including ElevenLabs and Gemini Pro, and enables the first zero-shot voice cloning capability for Lao. Audio Samples are available at: https://luoji.cn/static/multilantts-demo-main/.

## 1. Introduction

Conventional text-to-speech (TTS) systems typically rely on grapheme-to-phoneme (G2P) conversion, which creates a significant bottleneck for languages with intricate phonological rules and irregular scripts (Ren et al., 2021; Shen et al., 2018). Spoken Language Models (SLMs) have emerged as a powerful alternative by modeling discretized neural tokens in an autoregressive manner, thereby bypassing explicit G2P modules and enabling advanced capabilities (Wang et al., 2023; Borsos et al., 2023; Kharitonov et al., 2022). Nevertheless, this transition from rule-based pipelines to data-driven modeling does not eliminate the global resource disparity. Because SLMs remain fundamentally constrained by the volume and quality of transcribed speech, their performance often degrades when applied to languages outside the major high-resource groups such as English or Mandarin (Pratap et al., 2024; Grützner-Zahn et al., 2024).

We identify Southeast Asia as a critical domain for evaluating low-resource SLMs, as it highlights a stark contrast to the data abundance of mainstream languages(Nguyen et al., 2024; Susanto et al., 2025). Within this region, we categorize languages along a resource spectrum to address distinct modeling challenges. Thai represents a phonetically complex but digitally under-represented language, where its five lexical tones and complex tonal sandhi require precise modeling(Geng et al., 2025; Shen et al., 2024). In contrast, Lao exemplifies an even more constrained regime characterized by severe data scarcity. While authentic Lao corpora are not entirely absent, their publicly available volume is exceptionally limited, which is further compounded by a smaller speaker population and fewer commercial applications(Liu et al., 2025a; McGiff & Nikolov, 2025).

A natural solution is to leverage synthetic data (de Gibert et al., 2025; Ulm et al., 2025). Existing deterministic TTS systems can already synthesize speech for many under-represented languages with reasonable phonetic accuracy, even if the outputs are prosodically flat (Shumailov et al., 2023). This motivates our approach of fine-tuning pre-trained SLM backbones with synthetic data, where the backbone contributes expressive prosodic priors while synthetic data ensures phonetic stability (Minixhofer et al., 2025; Kwon et al., 2025). However, scaling this paradigm

[1]Beijing University of Posts and Telecommunications, Beijing, China [2]Beijing Logic Intelligence Technology, Beijing, China [3]University of California, USA [4]Northwestern University, USA [5]Eastern Institute of Technology, Ningbo, China. Correspondence to: xiaoyushen <xyshen@eitech.edu.cn >.

*Proceedings of the 43rd International Conference on Machine Learning*, Seoul, South Korea. PMLR 306, 2026. Copyright 2026 by the author(s).

on Thai reveals a critical trade-off that we define as the Stability-Expressivity Gap. We observe that increasing the synthetic data ratio improves phonetic stability but progressively degrades prosodic naturalness. Through scaling law analysis, we discover that beyond a critical synthetic ratio, the model's prosodic distribution collapses toward the low-entropy patterns of synthetic data, a phenomenon we call Synthetic Erosion (Alemohammad et al., 2023; Radford et al., 2023).

This scaling behavior locks practitioners into suboptimal data configurations. To break this constraint, we propose Disentanglement-Guided Self-Alignment (DGSA), a self-alignment framework that exploits the architectural properties of Flow-Matching SLMs (Du et al., 2024; Zhang et al., 2025b). Our key observation is that these models separate prosody from timbre. The Text-Speech LM encodes content and speaking style via optional style tokens, while the Flow-Matching Transformer independently controls speaker identity via timbre embeddings. By selectively enabling style conditioning, we generate outputs with identical speaker identity but contrasting prosodic quality. This architectural disentanglement enables the model to construct its own preference pairs without external annotation, achieving self-supervised alignment that simultaneously improves stability and expressivity.

DGSA relies on real speech to extract style references. However, in extreme scenarios where high-quality authentic corpora are practically inaccessible, the autoregressive process becomes unstable without authentic references. This lack of human-recorded anchors often causes token predictions to collapse into repetitive loops or phonetic hallucinations (Zhou et al., 2024a). To address this, we introduce Temperature-Driven Self-Critique (TDSC), a closed-loop mechanism that generates candidates across temperature gradients, applies ASR-based filtering, and iteratively refines the model using accepted samples as pseudo-real anchors (Kahn et al., 2020; Yuan et al., 2024). This self-improvement loop stabilizes decoding while recovering prosodic diversity without the need for human-labeled data.

Our main contributions are:

- **Characterizing the Stability-Expressivity Gap.** We conduct the first systematic scaling study of synthetic data in low-resource SLMs, revealing a non-monotonic trade-off and identifying the Synthetic Erosion phenomenon through multi-metric evaluation.

- **Disentanglement-Guided Self-Alignment (DGSA).** We discover that the prosody-timbre separation in Flow-Matching SLMs enables self-contrastive preference construction, achieving annotation-free alignment that simultaneously improves stability and expressivity.

- **Temperature-Driven Self-Critique (TDSC).** We in-

troduce a closed-loop self-refinement mechanism for low-resource languages, which stabilizes autoregressive decoding through temperature-guided exploration and linguistic filtering.

Evaluations on Thai and Lao achieve state-of-the-art performance. Our Thai system surpasses commercial APIs including ElevenLabs in zero-shot voice cloning; our Lao system represents the first TTS capable of voice cloning for this language. These results demonstrate a promising pathway for high-fidelity synthesis in low-resource languages where baseline ASR capability is available.

**Conflict of Interest Disclosure.** One author is affiliated with Beijing Logic Intelligence Technology. This work evaluates the authors' research system against several commercial TTS APIs. The authors declare no other financial conflicts of interest related to this paper.

## 2. Stability-Expressivity Gap

Given the scarcity of natural recordings, training spoken language models for low-resource languages often necessitates synthetic data augmentation (Huybrechts et al., 2021; Du & Yu, 2020; Ragni et al., 2014; Rosenberg et al., 2019). However, as we demonstrate empirically in this section, the benefit of synthetic data is bounded: beyond a critical ratio, continued scaling induces systematic degradation of the model's output diversity—a phenomenon we term Synthetic Erosion (Shumailov et al., 2024; Minixhofer et al., 2025; Alemohammad et al., 2023).

**Synthetic Data Construction.** We emphasize that all synthetic data in our framework are standard text-speech pairs $(x, y_{\text{syn}})$. The text $x$ is sourced from external corpora (multilingual C4), entirely disjoint from any real speech transcripts. The speech $y_{\text{syn}}$ is generated by off-the-shelf open-source TTS models (MMS-TTS, Seamless-M4T-v2, Typhoon2-Audio) that were *not* trained on any of our data. Within the SLM training pipeline, both real and synthetic speech are mapped by the same speech tokenizer (S3Tokenizer) into a common discrete token space, and the model is trained on the resulting conditional distribution $\pi_\theta(y|x)$. Since deterministic TTS typically produces flatter token distributions than human speech, the mixed distribution $p_\alpha$ admits the concavity-based analysis below at the token level. This cross-modal construction differs from prior synthetic data studies in the image/text domain (Shumailov et al., 2023; Alemohammad et al., 2023), where synthetic erosion arises from iterative self-consumption; in our setting, the erosion stems from the low-entropy bias of external TTS outputs rather than recursive self-generation. For Thai, training data and the TSynC-2 (Wutiwiwatchai et al.) test set are strictly disjoint—no utterances, speakers, or tran-

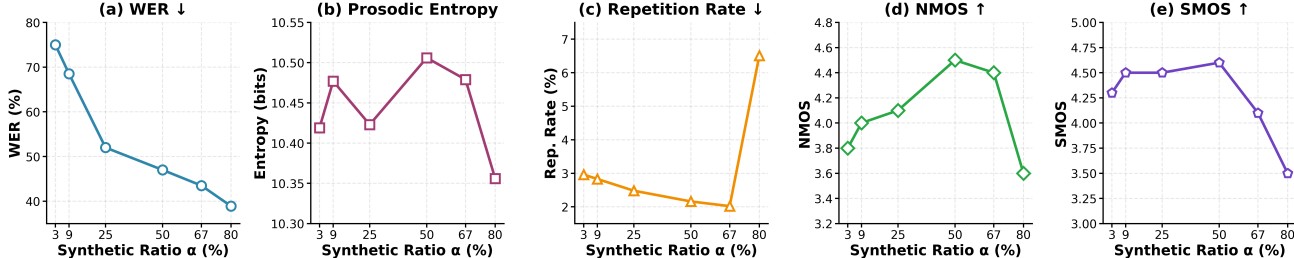

*Figure 1.* Scaling behavior of objective metrics as synthetic data ratio $\alpha$ increases. WER decreases monotonically, indicating improved stability. In contrast, token entropy $H_p$, repetition rate, NMOS, and SMOS exhibit non-monotonic trends—peaking around $\alpha \approx 50\%$ before degrading, revealing the Synthetic Erosion phenomenon. Notably, $H_p$ tracks NMOS, supporting its use as a lightweight proxy for prosodic diversity.

scripts are shared. Detailed data statistics are provided in Appendix F.

**Problem Formulation.** We consider autoregressive speech synthesis under mixed supervision. Let $\pi_\theta$ denote a policy that generates a sequence of discrete speech tokens $y \in \mathcal{V}^T$ conditioned on input text $x$ (Lakhotia et al., 2021; Wang et al., 2023):

$$\pi_\theta(y \mid x) = \prod_{t=1}^{T} \pi_\theta(y_t \mid y_{<t}, x) \qquad (1)$$

The training corpus $\mathcal{D} = \mathcal{D}_{\text{real}} \cup \mathcal{D}_{\text{syn}}$ consists of pairs $(x, y)$ combining authentic human recordings $\mathcal{D}_{\text{real}}$ with synthetic speech $\mathcal{D}_{\text{syn}}$. We parameterize the data composition by the synthetic ratio $\alpha = |\mathcal{D}_{\text{syn}}|/|\mathcal{D}|$ and train via maximum likelihood:

$$\mathcal{L}(\theta) = -\mathbb{E}_{(x,y) \sim \mathcal{D}}\big[\log \pi_\theta(y \mid x)\big] \qquad (2)$$

For low-resource languages such as Thai and Lao, data scarcity often necessitates $\alpha > 0.8$ (Rosenberg et al., 2019; Thai et al., 2019). The central question is: how does $\alpha$ affect the expressivity of the learned policy.

**Token Entropy as a Diagnostic Signal.** The gold standard for evaluating prosodic quality is human judgment, typically measured via Naturalness Mean Opinion Score (NMOS) (Streijl et al., 2016). However, NMOS evaluation is expensive and cannot be computed during training. We therefore seek a lightweight, automatically computable proxy that correlates with perceptual naturalness.

We propose monitoring token-level entropy as prosodic entropy over generated utterances:

$$H_p = -\sum_{v \in \mathcal{V}} p(v) \log p(v) \qquad (3)$$

where $p(v)$ is the empirical frequency of token $v$ across all samples. The motivation for this choice stems from the architectural design of modern Flow-Matching

SLMs (Mehta et al., 2024; Le et al., 2023). As established by CosyVoice (Du et al., 2024) and Vevo (Zhang et al., 2025b), autoregressive tokens encode content and prosody (pitch contours, rhythm, emphasis), while the Flow-Matching decoder separately controls speaker timbre via independent embeddings (Ju et al., 2024). This architectural decoupling implies that token-level statistics primarily reflect prosodic variation rather than speaker identity.

We emphasize that $H_p$ is a diagnostic signal for distributional diversity. Its validity as a proxy is established empirically: as shown in Figure 1 and detailed in Section 5.2, $H_p$ exhibits the same non-monotonic trend as NMOS across all synthetic ratios, with both metrics peaking near $\alpha \approx 50\%$. This tight correspondence—observed consistently across experimental conditions—supports the use of $H_p$ as a lightweight indicator of prosodic richness when human evaluation is impractical.

**Empirical Characterization of Synthetic Erosion.** We train models with fixed real data while varying synthetic data across $\alpha \in \{3\%, 9\%, 25\%, 50\%, 67\%, 80\%\}$, and observe a consistent non-monotonic pattern (Figure 1): WER decreases monotonically with $\alpha$, yet $H_p$, repetition rate, NMOS, and SMOS all peak near $\alpha \approx 50\%$ before degrading. This reveals that stability and expressivity decouple beyond a critical ratio—improving one necessarily sacrifices the other under naive data scaling.

To provide intuition for this behavior, we model the effective training distribution as a mixture:

$$p_\alpha = (1 - \alpha) \cdot p_{\text{real}} + \alpha \cdot p_{\text{syn}} \qquad (4)$$

Deterministic TTS engines produce outputs with substantially lower variation than human speech (Ren et al., 2021), i.e., $H(p_{\text{syn}}) < H(p_{\text{real}}) - \delta$ for some $\delta > 0$. A classical property of mixture distributions is that $H(p_\alpha)$ is strictly concave in $\alpha$ when $p_{\text{real}} \neq p_{\text{syn}}$, implying the existence of a unique peak $\alpha^*$ with the two-phase structure (see Ap-

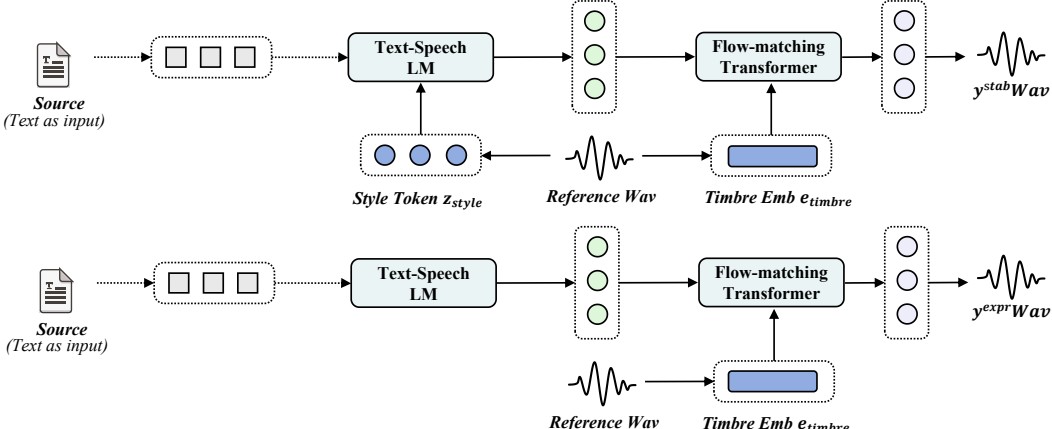

*Figure 2.* **Disentanglement-Guided Self-Alignment (DGSA).** Flow-Matching SLMs separate prosody (Text-Speech LM) from timbre (Flow-Matching Transformer). Enabling the style token produces expressive output $y^{\text{expr}}$; disabling it yields stable but flat output $y^{\text{stab}}$. DGSA aligns both toward real speech $y^{\text{real}}$ via dual preference objectives.

pendix B for derivation):

$$\frac{dH(\alpha)}{d\alpha} \begin{cases} > 0 & \alpha < \alpha^* \quad \textit{(Diversity Increase)} \\ < 0 & \alpha > \alpha^* \quad \textit{(Synthetic Erosion)} \end{cases} \quad (5)$$

At low $\alpha$, synthetic data introduces token patterns absent from limited real data, increasing overall diversity. Beyond $\alpha^*$, the low-entropy synthetic distribution dominates and diversity monotonically decreases. We stress that this mixture analysis serves as a qualitative explanatory framework: it describes the training data distribution rather than the learned model's output, and the actual $\alpha^*$ is dataset-specific. Nevertheless, the predicted non-monotonicity aligns closely with our empirical observations, providing useful intuition for why naive scaling fails and motivating the alignment-based corrections we propose next.

## 3. Disentanglement-Guided Self-Alignment

The scaling behavior in Section 2 locks practitioners into suboptimal configurations: reducing $\alpha$ sacrifices phonetic coverage (Le et al., 2023), while increasing it induces Synthetic Erosion (Shumailov et al., 2023). We break this constraint through a self-alignment framework that exploits the architectural properties of Flow-Matching SLMs (Mehta et al., 2024; Guan et al., 2025), enabling preference-based correction without human annotation (Liu et al., 2025b; Zhang et al., 2025a; Zhou et al., 2024b).

**Architectural Foundation.** Flow-Matching SLMs decompose generation into two independent pathways (Du et al., 2024; Zhang et al., 2025b): the Text-Speech LM produces tokens encoding content and prosody, optionally condi-

tioned on a style prefix $z_{\text{style}}$; the Flow-Matching Transformer converts tokens to waveforms using timbre embeddings $e_{\text{timbre}}$. These signals operate independently—we can toggle prosodic guidance while preserving speaker identity, enabling controlled generation of outputs that isolate specific attributes.

**Dual-Mode Generation.** For text $x$ with real speech $y^{\text{real}}$, we generate two complementary outputs:

$$y^{\text{expr}} = \pi_\theta(x \mid z_{\text{style}}, e_{\text{timbre}}), \quad (6)$$

$$y^{\text{stab}} = \pi_\theta(x \mid \varnothing, e_{\text{timbre}}). \quad (7)$$

The expressive output $y^{\text{expr}}$ inherits prosodic variation but may accumulate phonetic errors; the stable output $y^{\text{stab}}$ is phonetically consistent but prosodically flat.

**Dual-Objective Alignment.** Real speech exhibits both stability and expressivity. We construct two preference sets aligning each mode toward $y^{\text{real}}$:

$$\mathcal{T}_{\text{stab}} = \{(x, y^{\text{real}}, y^{\text{expr}})\}, \quad (8)$$

$$\mathcal{T}_{\text{expr}} = \{(x, y^{\text{real}}, y^{\text{stab}})\}. \quad (9)$$

$\mathcal{T}_{\text{stab}}$ teaches that real speech is preferred over expressive-but-erroneous outputs; $\mathcal{T}_{\text{expr}}$ teaches that real speech is preferred over stable-but-flat outputs. Both share $y^{\text{real}}$ as the positive example but target different failure modes.

The combined loss is:

$$\mathcal{L}_{\text{DGSA}} = \lambda_s \mathcal{L}_{\text{DPO}}(\mathcal{T}_{\text{stab}}) + \lambda_e \mathcal{L}_{\text{DPO}}(\mathcal{T}_{\text{expr}}), \quad (10)$$

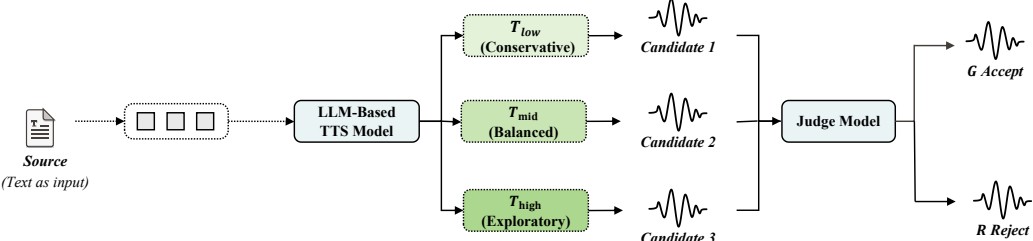

*Figure 3.* **Temperature-Driven Self-Critique (TDSC)** For each input, the model generates candidates at temperatures $T \in \{low, mid, high\}$, spanning conservative (stable) to exploratory (expressive) outputs. The Judge Model filters candidates by WER, length, and repetition criteria, yielding accepted ($\mathcal{G}$) and rejected ($\mathcal{R}$) sets for preference-based refinement.

where each DPO term follows the standard formulation (Rafailov et al., 2023):

$$\mathcal{L}_{\text{DPO}}(\mathcal{T}) = -\mathbb{E}_{(x,y^+,y^-)}\big[\log \sigma(\beta\,\Delta_\theta)\big], \quad (11)$$

$$\Delta_\theta = \log \frac{\pi_\theta(y^+|x)}{\pi_{\text{ref}}(y^+|x)} - \log \frac{\pi_\theta(y^-|x)}{\pi_{\text{ref}}(y^-|x)}. \quad (12)$$

Here $\pi_{\text{ref}}$ is the frozen SFT policy.

**Stage Separation.** We emphasize that the training pipeline is strictly sequential. **Stage 1 (SFT):** The model is fine-tuned on the mixed corpus $\mathcal{D}_{\text{real}} \cup \mathcal{D}_{\text{syn}}$ via maximum likelihood (Eq. 2). **Stage 2 (Generation):** The SFT checkpoint is *frozen*, and $y^{\text{expr}}, y^{\text{stab}}$ are generated from this frozen model. These outputs never enter the SFT objective. **Stage 3 (DGSA Alignment):** DPO is applied on the constructed preference triplets starting from the same SFT checkpoint. This design ensures that the SFT baseline and the DGSA model share exactly the same Stage-1 training; the comparison in Section 5.3 isolates the effect of alignment alone.

**Dynamic Weight Scheduling.** To counter synthetic erosion without destabilizing training, we employ a dynamic crossover schedule determined by the critical ratio $\alpha^*$. Below this threshold, we prioritize stability ($\lambda_s = 1$). Beyond it, we linearly ramp up the expressivity weight $\lambda_e$ proportional to the excess synthetic data, while correspondingly reducing $\lambda_s$ to rebalance the objective:

$$\lambda_e = \max\left(0, \frac{\alpha - \alpha^*}{1 - \alpha^*}\right), \quad (13)$$

$$\lambda_s = 1 - \lambda_e. \quad (14)$$

This mechanism (Figure 5) ensures that corrective pressure for prosodic diversity increases precisely when the training distribution becomes dominated by synthetic data.

## 4. Temperature-Driven Self-Critique

DGSA requires real speech recordings to anchor preference construction. For low-resource languages such as Lao where authentic corpora are practically inaccessible(Pratap et al., 2024; Gong et al., 2024), we introduce Temperature-Driven Self-Critique (TDSC), a closed-loop mechanism that bootstraps policy refinement from model self-evaluation alone.

**Multi-Temperature Trajectory Exploration.** The sampling temperature $T$ controls the entropy of autoregressive decoding(Holtzman et al., 2019). We define the temperature-scaled policy as:

$$\pi_\theta^{(T)}(y_t \mid y_{<t}, x) \propto \pi_\theta(y_t \mid y_{<t}, x)^{1/T} \quad (15)$$

Low temperatures ($T < 1$) yield stable but monotonous outputs; high temperatures ($T > 1$) enable prosodic diversity at the risk of phonetic errors(Wang et al., 2023; Mayer et al., 2025; Ju et al., 2024; Zhang et al., 2024b). Rather than committing to a single $T$, TDSC generates multiple candidates across a temperature gradient $\mathcal{T} = \{T_{\text{low}}, T_{\text{mid}}, T_{\text{high}}\}$ for each input text $x$, producing a diverse candidate pool that spans the Stability-Expressivity spectrum(Mayer et al., 2025; Wu & Tambe, 2025).

**Self-Critique and Pair Construction.** Lacking ground-truth references, we construct a composite judge to evaluate candidate $y$ against input $x$. We define three strict criteria:

$$\mathcal{C}(y) = \begin{cases} 1 & \text{if } \begin{cases} \texttt{WER}(y) < \tau_w \\ \texttt{Rep}(y) < \tau_r \\ \texttt{Len}(y) \in [\gamma_{\min}|x|, \gamma_{\max}|x|] \end{cases} \\ 0 & \text{otherwise} \end{cases} \quad (16)$$

where $\texttt{Rep}(y)$ denotes the repetition rate, defined as the fraction of positions in the token sequence where $k+1$ consecutive identical tokens appear (we set $k = 4$ to capture

persistent loops rather than natural phonetic gemination; see Appendix E for the formal definition). The length bounds are dynamically scaled by the text length $|x|$ to reject distinct duration failures while accommodating natural speech rate variations. Based on $\mathcal{C}(y)$, we mine preference pairs $(y_w, y_l)$ to construct the training sets. The accepted set $\mathcal{G}^{(k)}$ consists of candidates that satisfy $\mathcal{C}(y) = 1$, from which we select the sample with the lowest WER as the winner $y_w$. Crucially, to prevent the model from exploiting length heuristics during DPO, the rejected sample $y_l$ is selected from candidates that pass the length and repetition filters but exhibit high WER. This ensures the optimization focuses on phonetic accuracy rather than duration artifacts.

**Recursive Refinement.** TDSC operates as an iterative closed-loop. In each iteration $k$, we refine the policy $\pi_\theta$ through a two-stage optimization process using the filtered datasets. First, we stabilize the model by maximizing the likelihood of high-quality samples in $\mathcal{G}^{(k)}$ via Supervised Fine-Tuning (SFT):

$$\mathcal{L}_{\text{SFT}}(\theta) = -\mathbb{E}_{y \in \mathcal{G}^{(k)}} \left[ \log \pi_\theta(y \mid x) \right] \tag{17}$$

Subsequently, to improve discrimination, we apply Direct Preference Optimization (DPO) using pairs $(y_w, y_l)$ constructed from $\mathcal{G}^{(k)}$ and the rejected set $\mathcal{R}^{(k)}$:

$$\mathcal{L}_{\text{DPO}}(\theta) = -\mathbb{E}_{(y_w, y_l)} \left[ \log \sigma(\beta \, \Delta_\theta) \right], \tag{18}$$

$$\Delta_\theta = \log \frac{\pi_\theta(y_w|x)}{\pi_{\text{ref}}(y_w|x)} - \log \frac{\pi_\theta(y_l|x)}{\pi_{\text{ref}}(y_l|x)}. \tag{19}$$

This sequential update ensures the model first consolidates its ability to generate stable speech (SFT), then learns to suppress specific failure modes like hallucinations (DPO). As the policy stabilizes, we expand the exploration space by increasing the temperature limit $T_{high}^{(k)} = T_{high}^{(0)} + \gamma \cdot k$, progressively recovering prosodic diversity.

## 5. Experiments

### 5.1. Experimental Setup

We evaluate our framework on Thai and Lao. Training data for Thai consists of 300h of ASR-filtered real speech from Common Voice and 1,200h of synthetic data, while Lao relies entirely on 1,500h of synthetic speech. We compare our system against open-source baselines (PythaiTTS(Phatthiyaphaibun et al., 2023), Typhoon2-Audio(Pipatanakul et al., 2024), Seamless-M4T-v2(Barrault et al., 2023), MMS-TTS(Pratap et al., 2024)) and commercial APIs (Gemini, Azure, ElevenLabs v3). To ensure reproducibility, all commercial evaluations were frozen on January 25, 2025, utilizing the widely recognized TSynC-2 (Wutiwiwatchai et al.) (Thai) and Common Voice(Ardila et al., 2020) (Lao) corpora as benchmarks for evaluation.

*Table 1.* Scaling behavior across synthetic data ratios. Best expressivity metrics ($H_p$, NMOS, SMOS) occur at $\alpha \approx 50\%$. The $\alpha = 100\%$ row (pure synthetic, 0h real) confirms severe Synthetic Erosion. Subjective metrics are reported with 95% CI.

| Syn. (hours) | $\alpha$ (%) | WER↓ (%) | $H_p$ (bits) | Rep.↓ (%) | NMOS↑ (Mean ± CI) | SMOS↑ (Mean ± CI) |
|---|---|---|---|---|---|---|
| 10 | 3 | 75.0 | 10.419 | 2.96 | $3.82 \pm 0.09$ | $4.31 \pm 0.08$ |
| 30 | 9 | 68.5 | 10.477 | 2.83 | $4.01 \pm 0.08$ | $4.52 \pm 0.07$ |
| 53 | 15 | 58.2 | 10.455 | 2.62 | $4.08 \pm 0.08$ | $4.54 \pm 0.06$ |
| 100 | 25 | 52.0 | 10.423 | 2.48 | $4.13 \pm 0.07$ | $4.55 \pm 0.06$ |
| 200 | 40 | 49.2 | 10.498 | 2.21 | $4.38 \pm 0.07$ | $4.60 \pm 0.06$ |
| 300 | 50 | 47.0 | **10.506** | 2.16 | $\mathbf{4.51 \pm 0.06}$ | $\mathbf{4.63 \pm 0.05}$ |
| 450 | 60 | 44.8 | 10.491 | 2.08 | $4.45 \pm 0.06$ | $4.35 \pm 0.07$ |
| 600 | 67 | 43.5 | 10.479 | **2.02** | $4.42 \pm 0.07$ | $4.12 \pm 0.08$ |
| 1200 | 80 | 38.9 | 10.356 | 6.51 | $3.61 \pm 0.09$ | $3.54 \pm 0.10$ |
| 1500 | 100 | **36.2** | 10.210 | 9.83 | $3.08 \pm 0.10$ | $3.01 \pm 0.11$ |

The model architecture builds on CosyVoice 2(Du et al., 2024). Performance is evaluated using objective metrics (WER, Speaker Similarity, Prosodic Entropy) and subjective Mean Opinion Score (MOS) tests. For WER calculation and data filtering, we employ distinct ASR models to ensure robust accuracy: for Thai, we utilize Whisper-large-v3 (Radford et al., 2023); for Lao, we adopt Dolphin-small(Meng et al., 2025), as empirical testing revealed it significantly outperforms Whisper in recognizing Lao. Subjective evaluation includes Naturalness MOS (NMOS) to assess prosody and Speaker Similarity MOS (SMOS) to measure identity preservation (Streijl et al., 2016). The evaluation follows a double-blind, randomized, within-subject design involving 20 native speakers per language. We report metrics with 95% Confidence Intervals (CI) and conduct paired $t$-tests to verify statistical significance ($p < 0.05$). Detailed hyperparameters, including the training infrastructure, filtering thresholds for TDSC and model configurations, are provided in Appendix D.

### 5.2. Scaling Experiments

We first examine how the synthetic data ratio $\alpha$ affects model behavior, validating the non-monotonic pattern discussed in Section 2. We train models with fixed 300h real speech while varying synthetic data from 10h to 1,500h, corresponding to synthetic ratios $\alpha \in \{3\%, 9\%, 15\%, 25\%, 40\%, 50\%, 60\%, 67\%, 80\%, 100\%\}$.

**Phase I: Diversity Increase ($\alpha < 50\%$).** As shown in Table 1 and Figure 1, increasing synthetic data initially benefits both stability and expressivity. WER drops from 75% to 47% as phonetically consistent synthetic supervision regularizes training. Concurrently, token entropy $H_p$ rises from 10.42 to 10.51 bits, and repetition rate decreases from 2.96% to 2.16%, indicating that the model escapes underfitting and explores richer output trajectories. Subjective scores confirm this trend: NMOS improves from 3.8 to 4.5, and SMOS

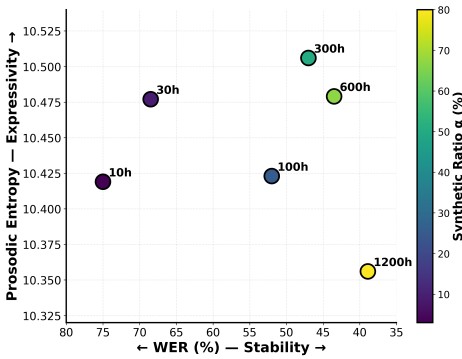

*Figure 4.* Stability-Expressivity trade-off space. Each point represents a model trained with different synthetic data ratios. Lower WER (rightward) indicates better stability; higher $H_p$ (upward) indicates better expressivity. The 300h configuration achieves the best balance, while excessive synthetic data (1200h, 1500h) sacrifices expressivity for marginal stability gains.

*Table 2.* Controlled comparison of high vs. low $H_p$ samples paired by identical text and matched WER. Improvements in acoustic features and perceived expressivity ($p < 0.01$) validate $H_p$ as a prosodic indicator.

| Metric | High $H_p$ | Low $H_p$ | $\Delta$ |
|---|---|---|---|
| WER (%) ↓ | 38.2 | 38.8 | –0.6 (n.s.) |
| F0 Std. (Hz) ↑ | 42.6 | 35.8 | +6.8 |
| F0 Range (Hz) ↑ | 128.4 | 109.7 | +18.7 |
| F0 Corr. ↑ | 0.68 | 0.52 | +0.16 |
| F0 RMSE (Hz) ↓ | 28.3 | 39.1 | –10.8 |
| Energy Std. (dB) ↑ | 7.4 | 6.1 | +1.3 |
| *Expressivity MOS* ↑ | 4.2 | 3.7 | +0.5 |

from 4.3 to 4.6.

**Phase II: Synthetic Erosion ($\alpha > 50\%$).** Beyond $\alpha \approx 50\%$, a striking divergence emerges (Figure 4). WER continues to improve ($47\% \rightarrow 36\%$), yet all expressivity metrics degrade. Token entropy $H_p$ decays from 10.51 to 10.21 bits at $\alpha = 100\%$, consistent with the two-phase structure in Eq. (5). Most dramatically, repetition rate surges from 2.02% to 9.83% at $\alpha = 100\%$, signaling distributional collapse toward repetitive patterns. Subjective quality suffers correspondingly: NMOS drops from 4.5 to 3.1, and SMOS from 4.6 to 3.0—both falling well below the $\alpha = 3\%$ baseline despite superior WER. Notably, the denser scaling points reveal that SMOS degrades earlier than NMOS ($\alpha = 60\%$: SMOS 4.35 vs. NMOS 4.45), suggesting that speaker identity preservation is more sensitive to synthetic dominance than prosodic naturalness. The pure-synthetic regime ($\alpha = 100\%$) is consistent with the Lao results (Table 4, NMOS = 3.12, $H_p$ = 10.08), providing cross-language validation that Synthetic Erosion is a general phenomenon rather than a Thai-specific artifact.

*Table 3.* Alignment methods comparison at $\alpha = 80\%$. DGSA simultaneously achieves high expressivity and stability. Intervals denote 95% CI.

| Method | WER↓ | $H_p$↑ | Rep.↓ | NMOS↑ | SMOS↑ |
|---|---|---|---|---|---|
| SFT Baseline | **38.9** | 10.36 | 6.51 | $3.61 \pm 0.09$ | $3.54 \pm 0.10$ |
| Standard DPO | 45.2 | 10.49 | 4.08 | $3.92 \pm 0.08$ | $3.81 \pm 0.09$ |
| Rejection Sampling | 40.5 | 10.41 | 5.18 | $3.75 \pm 0.08$ | $3.66 \pm 0.09$ |
| **DGSA (Ours)** | **38.9** | **10.52** | **2.82** | $\mathbf{4.42 \pm 0.07}$ | $\mathbf{4.53 \pm 0.06}$ |

**Validating Token Entropy as a Prosodic Indicator.** We validate $H_p$ as a proxy for prosody based on the architectural decoupling in Flow-Matching SLMs, where AR tokens encode prosody while the decoder handles timbre (Zhang et al., 2025b; Du et al., 2024). To isolate prosodic variation from stability and content, we conduct a controlled study using 2,000 sample pairs. Crucially, each pair is generated from the identical text and matched for intelligibility (WER $\in [35\%, 42\%]$), but contrasted by entropy ($H_p$ top vs. bottom quartile). As shown in Table 2, high-$H_p$ samples exhibit richer pitch dynamics (F0 range/std), stronger F0 correlation with ground-truth recordings, and greater energy variation despite comparable WER. Higher human ratings (4.2 vs. 3.7 MOS) confirm that $H_p$ captures perceptually meaningful expressivity rather than generation noise.

### 5.3. DGSA Evaluation

We evaluate DGSA at $\alpha = 80\%$, where Synthetic Erosion is most severe. We compare against the SFT Baseline, Standard DPO (single expressivity objective), and Rejection Sampling (inference-time filtering).

**Main Results.** Table 3 demonstrates that DGSA successfully closes the Stability-Expressivity Gap. It maintains the rigorous stability of the SFT baseline (WER: 38.9% for both) while substantially recovering expressivity ($H_p$ improves from 10.36 to 10.52 bits; NMOS from 3.6 to 4.4). In contrast, Standard DPO improves $H_p$ but degrades WER to 45.2%, confirming that single-objective alignment without architectural guidance sacrifices phonetic accuracy. Rejection Sampling provides only marginal gains over SFT. By effectively decoupling the two objectives, DGSA achieves the best trade-off, combining the stability of supervised learning with rich prosodic diversity.

**Why $\alpha = 80\%$?** By design, DGSA applies zero correction at $\alpha \leq \alpha^*$. Our dynamic scheduling (Eq. 14) gives $\lambda_e = 0$ when $\alpha = 50\%$, so DGSA reduces exactly to SFT at this ratio. This is intentional: Synthetic Erosion has not yet emerged at $\alpha = 50\%$, so no corrective pressure is needed. DGSA targets the high-$\alpha$ regime (70–90%) where practitioners are forced by data scarcity—here the model has better pronunciation (WER 38.9% vs. 47.0% at $\alpha = 50\%$) but suffers expressivity collapse (NMOS 3.61 vs. 4.51), which DGSA restores to 4.42.

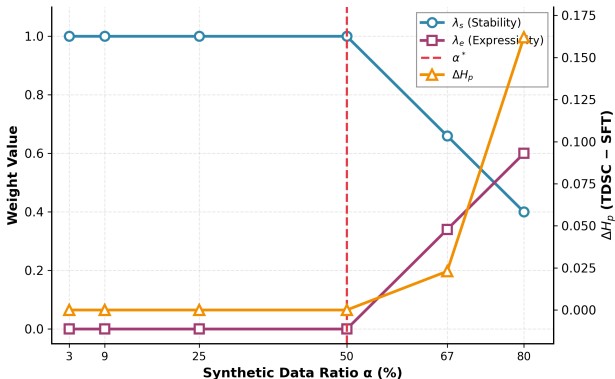

*Figure 5.* Dynamic weight scheduling and $H_p$ recovery. Below $\alpha^* = 50\%$, $\lambda_e = 0$ (no correction needed). Beyond $\alpha^*$, $\lambda_e$ activates and $\Delta H_p$ scales proportionally.

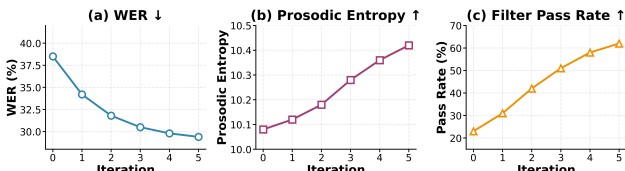

*Figure 6.* TDSC iteration dynamics over 5 refinement rounds. WER decreases steadily while $H_p$ rises in later iterations as $T_{\max}$ expands. Pass rate increases from 23% to 62%, indicating progressive quality improvement.

**Ablation Study.** Full results (Appendix C.1) confirm the Expressivity Objective is paramount; its removal yields flat outputs with the largest quality drop ($\Delta$NMOS = $-0.7$). Conversely, omitting the Stability Objective causes a spike in WER. Crucially, replacing Identity-Consistent Pairs with random pairing degrades both objectives, validating the necessity of prosody-timbre disentanglement.

**Dynamic Weight Behavior.** Figure 5 illustrates the adaptive trade-off in our scheduling. Below the critical threshold $\alpha^* = 50\%$, the system prioritizes pure stability ($\lambda_s = 1.0$), keeping the expressivity weight dormant ($\lambda_e = 0$). As synthetic erosion emerges beyond $\alpha^*$, the mechanism triggers a linear crossover: $\lambda_e$ scales up to inject prosodic diversity while $\lambda_s$ is correspondingly reduced. This rebalancing drives a sharp recovery in prosodic entropy ($\Delta H_p$), confirming that DGSA applies corrective pressure on-demand, strictly when the distribution begins to collapse.

## 5.4. TDSC Evaluation

We evaluate TDSC on Lao, a low-resource language with exceptionally limited real speech corpus. The model is trained entirely on 1,500h of synthetic data generated via cross-lingual transfer. We compare against alternative self-improvement strategies: Self-Training (iterative pseudo-labeling) and Rejection Sampling (inference-time filtering).

*Table 4.* TDSC evaluation on Lao. TDSC significantly outperforms alternative self-improvement methods ($p < 0.05$).

| Method | WER↓ (%) | $H_p$↑ (bits) | Rep.↓ (%) | NMOS↑ (Mean $\pm$ CI) |
|---|---|---|---|---|
| SFT Baseline | 38.5 | 10.08 | 7.62 | $3.12 \pm 0.09$ |
| Self-Training | 35.2 | 10.15 | 6.94 | $3.31 \pm 0.08$ |
| Rejection Sampling | 36.8 | 10.11 | 7.28 | $3.24 \pm 0.09$ |
| **TDSC (Ours)** | **29.8** | **10.42** | **4.15** | $\mathbf{3.94 \pm 0.07}$ |

**Main Results.** Table 4 compares TDSC against alternative self-improvement strategies. Starting from the same SFT baseline trained on synthetic data, TDSC achieves substantial gains: WER decreases by 24% relative (38.5% $\rightarrow$ 29.8%), repetition rate drops by 46% (7.62% $\rightarrow$ 4.15%), and NMOS improves by 0.8 points (3.1 $\rightarrow$ 3.9). In contrast, Self-Training provides only modest improvements (WER: 35.2%, NMOS: 3.3) and plateaus after few iterations due to confirmation bias. Rejection Sampling offers minimal benefit over SFT, as inference-time filtering cannot improve the underlying policy.

**Iteration Dynamics.** Figure 6 visualizes the closed-loop refinement process. WER decreases rapidly in early iterations (38.5% $\rightarrow$ 31.8% by $k = 2$) as the model learns from filtered high-quality samples, then converges gradually to 29.8%. Prosodic entropy $H_p$ exhibits a two-phase pattern: it remains stable during early iterations when $T_{\max}$ is conservative (0.8–1.0), then rises from 10.18 to 10.42 as the temperature curriculum expands to $T_{\max} = 1.3$, enabling greater prosodic exploration. The pass rate increases from 23% to 62%, confirming that generation quality improves with each iteration. This synchronized behavior validates the curriculum design: TDSC first establishes phonetic stability, then progressively recovers expressivity.

**Ablation Study.** Full ablation results (Appendix C.2) identify DPO loss as the primary driver of naturalness, causing the largest degradation ($\Delta$NMOS = $-0.5$) upon removal. Multi-Temperature exploration proves vital for prosodic richness ($H_p$), while the Temperature Curriculum prevents premature caps on expressivity. Finally, analysis confirms that our filter effectively aggregates complementary samples from stable and expressive regimes, balancing the trade-off.

## 5.5. Comparison with Existing Systems

We compare our full system against open-source and commercial TTS systems on both Thai and Lao. For Thai, we apply DGSA ($\alpha = 80\%$); for Lao, we use TDSC. Evaluation is conducted on held-out TSynC-2 for Thai, Common Voice for Lao. We consider two tasks: standard TTS common text-to-speech and zero-shot voice cloning (reproducing a target speaker from a short reference). For standard TTS, we

*Table 5.* Standard TTS comparison on Thai and Lao. Our system achieves the best expressivity (NMOS) with statistical significance ($p < 0.05$) against baselines.

| Language | Method | WER↓ (%) | NMOS↑ (Mean ± CI) |
|---|---|---|---|
| Thai | *Open-Source* | | |
| | PyThaiTTS | 78.4 | 2.91 ± 0.10 |
| | MMS-TTS | 53.5 | 3.24 ± 0.09 |
| | Typhoon2-Audio | 50.9 | 3.72 ± 0.08 |
| | Seamless-M4T-v2 | 47.8 | 3.55 ± 0.09 |
| | *Commercial* | | |
| | ElevenLabs-v3 | 40.6 | 4.21 ± 0.07 |
| | Gemini Flash | 40.2 | 3.93 ± 0.08 |
| | Gemini Pro | 41.9 | 4.05 ± 0.07 |
| | Microsoft Azure | **36.5** | 4.01 ± 0.07 |
| | **Ours (DGSA)** | 38.9 | **4.51 ± 0.06** |
| Lao | *Open-Source* | | |
| | MMS-TTS | 44.8 | 3.52 ± 0.09 |
| | *Commercial* | | |
| | Microsoft Azure | 41.8 | 3.91 ± 0.08 |
| | Gemini Flash | 34.2 | 4.12 ± 0.07 |
| | Gemini Pro | 35.6 | 4.10 ± 0.07 |
| | **Ours (TDSC)** | **29.8** | **4.53 ± 0.06** |

*Table 6.* Zero-shot voice cloning comparison. Our method outperforms baselines in speaker similarity and naturalness.

| Lang. | Method | WER↓ (%) | SIM↑ (0-1) | NMOS↑ (Mean ± CI) | SMOS↑ (Mean ± CI) |
|---|---|---|---|---|---|
| Thai | ElevenLabs-v3 | 42.3 | 0.78 | 4.21 ± 0.07 | 4.23 ± 0.07 |
| | **Ours (DGSA)** | **38.9** | **0.84** | **4.42 ± 0.06** | **4.51 ± 0.06** |
| Lao | Other Systems | | | *Not Supported* | |
| | **Ours (TDSC)** | **29.8** | **0.81** | **3.94 ± 0.07** | **4.32 ± 0.06** |

report WER and NMOS. For voice cloning, we additionally report speaker similarity (SIM) and speaker MOS (SMOS).

**Standard TTS Results.** Table 5 presents the standard TTS comparison. On Thai, our DGSA model achieves the highest NMOS (4.5), outperforming all commercial systems including ElevenLabs-v3 (NMOS: 4.2) and Azure TTS (NMOS: 4.0). While Azure achieves slightly lower WER (36.5% vs. 38.9%), this marginal stability gain comes at significant expressivity cost (NMOS 4.0 vs. 4.5), illustrating the Stability-Expressivity trade-off that our method addresses. The gap over open-source models is substantial: Typhoon2-Audio, the strongest open-source baseline, achieves only 50.9% WER and 3.7 NMOS.

On Lao, despite training with zero real speech, our TDSC model achieves both the lowest WER (29.8%) and highest NMOS (4.5)—surpassing the best commercial system (Gemini Flash) by 4.4% absolute WER and 0.4 NMOS points. The performance gap over MMS-TTS (44.8% WER, 3.5 NMOS), the only open-source system with Lao support, demonstrates the effectiveness of our synthetic-data framework combined with self-improvement techniques.

**Zero-Shot Voice Cloning Results.** Table 6 details performance on Thai and Lao. For Thai, our DGSA model outperforms ElevenLabs-v3, the only capable baseline, across all metrics by achieving superior intelligibility (WER 38.9%

vs. 42.3%) and speaker resemblance (SMOS 4.5 vs. 4.2). Regarding Lao, our system represents the only model capable of zero-shot cloning. Despite the heavy reliance on synthetic training data, TDSC achieves high fidelity, with a SMOS of 4.3 and a SIM of 0.81, which demonstrates that effective identity preservation is achievable without authentic target-language recordings.

## 6. Conclusion

In this work, we validate that fine-tuning expressive SLM backbones with flat synthetic data effectively extends high-fidelity synthesis to low-resource languages. Meanwhile, we demonstrate that this paradigm is constrained by a Stability-Expressivity Gap, where excessive synthetic ratios trigger Synthetic Erosion, a systematic collapse of the output distribution. To break this trade-off, we propose Disentanglement-Guided Self-Alignment (DGSA), which exploits the prosody-timbre separation in Flow-Matching SLMs to construct self-contrastive preference pairs. Furthermore, for low-resource Languages like Lao, we enable a robust pure-synthetic pipeline using Temperature-Driven Self-Critique (TDSC), stabilizing autoregressive decoding via temperature-guided exploration. Our approach achieves state-of-the-art results on Thai and Lao, surpassing commercial systems in zero-shot voice cloning.

## 7. Limitations

This work has several limitations. First, TDSC assumes the availability of a usable target-language ASR system for filtering generated samples. Although our Lao experiments show that moderate ASR quality is sufficient, languages without even basic ASR support may require unsupervised, cross-lingual, or weakly supervised recognition alternatives. Second, our experiments focus on Thai and Lao, two Southeast Asian tonal languages. While Synthetic Erosion is motivated by a general distributional mismatch between synthetic and human speech, broader evaluation on typologically diverse languages is needed to validate the generality of DGSA and TDSC. Third, TDSC introduces additional computation. This cost is non-trivial, but remains comparable to a large-scale SFT run and substantially cheaper than collecting native-speaker annotations for truly low-resource languages.

## Impact Statement

This research aims to bridge the digital divide for low-resource languages, specifically addressing the scarcity of high-quality speech technologies for Thai and Lao. By enabling high-fidelity synthesis and zero-shot adaptation without massive authentic corpora, our work contributes to linguistic inclusion and cultural preservation in the global AI landscape.

However, we acknowledge that the advancement of generative spoken language models, particularly the zero-shot voice cloning capabilities demonstrated in our experiments, carries inherent ethical risks. These technologies could potentially be misused for unauthorized impersonation, audio deepfakes, or telecommunications fraud.

To mitigate these risks, we emphasize that the methodologies and models presented in this paper are intended strictly for academic and educational purposes. We advocate for the responsible development of SLMs, where future deployment of such systems must be accompanied by robust safeguards, including:

- **Consent Protocols:** Ensuring voice cloning is performed only with the explicit permission of the speaker.
- **Anti-Spoofing Verification:** Developing countermeasures to detect synthetic artifacts in sensitive applications.

We believe that democratizing speech technology for the linguistic long-tail yields significant societal benefits, provided that the research community remains vigilant regarding misuse and actively contributes to safety mechanisms.

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

# A. Related Work

**Spoken Language Models and Disentangled Speech Representations.**    Spoken Language Models (SLMs) have emerged as a transformative paradigm by treating speech synthesis as conditional language modeling over discrete neural tokens(Borsos et al., 2023; Wang et al., 2023). Leveraging neural audio codecs and autoregressive Transformers, SLMs enable zero-shot voice cloning and in-context prosody adaptation without explicit linguistic front-ends(Défossez et al., 2023; Zeghidour et al., 2021). A crucial development in this area is the disentanglement of speech attributes into separate representational spaces(Zhang et al., 2024c; Li et al., 2025). Recent work such as Vevo (Zhang et al., 2025b) and CosyVoice (Du et al., 2024) demonstrates that well-designed SLM architectures can decouple *what to speak* (linguistic content), *how to speak* (prosody, including pitch, rhythm, and emphasis), and *who speaks* (speaker timbre)(Zhang et al., 2025b; Du et al., 2024). Specifically, the autoregressive transformer generates discrete tokens that encode content and prosodic style, while the Flow-Matching acoustic decoder conditions on separate timbre embeddings extracted from reference audio(Du et al., 2024; Peng et al., 2025). This architectural separation—where prosody is modeled in the discrete token space and timbre in the continuous acoustic space—provides the theoretical foundation for using token-level entropy as a measure of prosodic diversity(Wang et al., 2023; Zhang et al., 2024c). While these capabilities have been extensively validated in high-resource settings, their extension to low-resource languages faces fundamental challenges: the autoregressive generation process is highly sensitive to distributional shifts, and the absence of phonetically-transcribed corpora leads to severe acoustic hallucinations(Zhang et al., 2023; Bataev et al., 2025). Our work directly addresses this gap by analyzing and mitigating the pathologies that arise when SLMs are scaled to data-scarce languages.

**Synthetic Data Augmentation and Distributional Collapse.**    Synthetic data augmentation has become a standard practice for overcoming data scarcity in speech tasks(Jia et al., 2019; Minixhofer et al., 2025). Teacher-student distillation from high-quality TTS engines provides phonetically stable supervision, improving word error rates in both ASR and TTS systems(Ren et al., 2021). However, recent studies in the text domain have identified model collapse—a phenomenon where iterative training on synthetic outputs causes progressive degradation of distributional diversity(Shumailov et al., 2023; Alemohammad et al., 2023). While analogous effects have been hypothesized for speech, no prior work has systematically quantified how synthetic data ratios affect the prosodic expressivity of SLMs(Mizumoto et al., 2025; Minixhofer et al., 2025). We fill this gap by introducing Prosodic Entropy $H_p$ as a formal metric. This measure is well-motivated by the architectural disentanglement principle discussed above: since discrete tokens primarily encode prosodic rather than timbral information, the entropy of token distributions directly reflects the diversity of prosodic realizations(Zhang et al., 2024c; Wang et al., 2023). Our scaling law analysis characterizes Synthetic Erosion as a prosodic manifold collapse unique to the acoustic domain, distinct from the semantic degradation observed in text model collapse. We note that a key distinction from prior work is in the mechanism: text model collapse typically arises from iterative training on a model's own outputs, whereas our Synthetic Erosion results from training on low-entropy data generated by external, deterministic TTS systems.

**Preference Optimization for Speech Generation.**    Direct Preference Optimization (DPO) has emerged as a scalable alternative to reinforcement learning from human feedback (RLHF) for aligning generative models(Rafailov et al., 2023). In speech synthesis, preference-based methods have been applied to improve speaker similarity, emotional expressivity, and overall naturalness(Zhang et al., 2024a; Gao et al., 2025; Liu et al., 2025b). However, existing approaches typically optimize a single objective and assume access to high-quality human annotations for preference construction(Zhang et al., 2024a; 2025a). In low-resource settings, this assumption breaks down: the scarcity of native speakers and phonetic experts makes large-scale annotation infeasible, while the simultaneous requirements for linguistic stability and prosodic expressivity demand multi-objective optimization(Zhou et al., 2024b). Our proposed DGSA addresses both challenges by exploiting the architectural decoupling of timbre and prosody in Flow-Matching SLMs: by re-synthesizing the same content with different prosodic realizations while holding timbre constant, we construct preference triplets that isolate prosodic quality without requiring external annotation.

**Zero-shot Cross-lingual Transfer and Inference Stability.**    Zero-shot cross-lingual TTS aims to synthesize speech in unseen languages while preserving speaker identity from reference audio(Zhang et al., 2023; Le et al., 2023). While SLMs' shared multilingual token spaces enable such transfer, inference stability degrades significantly as target-language resources decrease(Zhang et al., 2023). Sampling strategies such as nucleus sampling and classifier-free guidance can mitigate surface-level artifacts but fail to address the fundamental Autoregressive Collapse that occurs when models lack exposure to authentic target-language distributions(Zheng & Maleki, 2025; Wang et al., 2024). Inspired by self-correction mechanisms in reasoning LLMs, we introduce Temperature-Driven Self-Critique (TDSC) to explore latent trajectories

across temperature scales and iteratively refine model behavior without human supervision(Madaan et al., 2023; Kumar et al., 2025).

## B. Derivation of Mixture Entropy Properties

This appendix provides supporting derivations for the mixture entropy analysis discussed in Section 2. The strict concavity of entropy over mixture distributions is a classical result; we include the details here for completeness and to establish notation.

### B.1. Setup

For distributions $p_{\text{real}}$ and $p_{\text{syn}}$ over a finite set $\mathcal{V}$ and mixing coefficient $\alpha \in [0,1]$, the mixture distribution is $p_\alpha = (1-\alpha)\,p_{\text{real}} + \alpha\,p_{\text{syn}}$. We write $H(\alpha) := H(p_\alpha)$ for the Shannon entropy of the mixture.

### B.2. Strict Concavity

**Lemma B.1.** *If $p_{real} \neq p_{syn}$, then $H(\alpha)$ is strictly concave on $[0,1]$.*

*Proof.* The entropy function $H : \Delta_{|\mathcal{V}|-1} \to \mathbb{R}$ is strictly concave on the probability simplex. For any $\alpha_1 \neq \alpha_2 \in [0,1]$ and $\lambda \in (0,1)$:

$$H(\lambda\alpha_1 + (1-\lambda)\alpha_2) = H\big(\lambda\,p_{\alpha_1} + (1-\lambda)\,p_{\alpha_2}\big) \tag{20}$$
$$> \lambda\,H(p_{\alpha_1}) + (1-\lambda)\,H(p_{\alpha_2}) \tag{21}$$

where the strict inequality uses $p_{\alpha_1} \neq p_{\alpha_2}$, which holds whenever $p_{\text{real}} \neq p_{\text{syn}}$. □

### B.3. Entropy Derivative

**Lemma B.2.** *The derivative of $H(\alpha)$ with respect to $\alpha$ is:*

$$\frac{dH(\alpha)}{d\alpha} = \sum_{v \in \mathcal{V}} \big(p_{real}(v) - p_{syn}(v)\big) \cdot \log \frac{1}{p_\alpha(v)} \tag{22}$$

*Proof.* Differentiating $H(\alpha) = -\sum_v p_\alpha(v) \log p_\alpha(v)$:

$$\frac{dH}{d\alpha} = -\sum_v \frac{\partial p_\alpha(v)}{\partial \alpha}\big(\log p_\alpha(v) + 1\big) \tag{23}$$
$$= -\sum_v \big(p_{\text{syn}}(v) - p_{\text{real}}(v)\big)\big(\log p_\alpha(v) + 1\big) \tag{24}$$

Since $\sum_v(p_{\text{syn}}(v) - p_{\text{real}}(v)) = 0$, the constant term vanishes, yielding Eq. (22). □

### B.4. Existence of a Unique Maximum

Combining the above: strict concavity (Lemma B.1) and continuity of $H(\alpha)$ on the compact interval $[0,1]$ guarantee the existence of a unique maximizer $\alpha^* \in (0,1)$, provided $H(\alpha)$ is not monotone. The latter is ensured when $H(p_{\text{syn}}) \neq H(p_{\text{real}})$, since $H(0) = H(p_{\text{real}})$ and $H(1) = H(p_{\text{syn}})$ differ while the strict concavity forces the function to lie above the chord connecting these endpoints. This completes the justification for the two-phase structure described in Section 2.

## C. Detailed Ablation Studies

This appendix provides detailed ablation studies for both DGSA (Section C.1) and TDSC (Section C.2), isolating the contribution of each component.

## C.1. DGSA Component Analysis

We evaluate DGSA at $\alpha = 80\%$, where Synthetic Erosion is most severe, by systematically removing each component. Table 7 presents the results on the Common Voice Thai test set.

*Table 7.* Ablation study of DGSA components at $\alpha = 80\%$. Each row removes one component from the full system. All components contribute to final performance; removing the Expressivity Objective causes the largest degradation ($\Delta$NMOS = $-0.7$).

| Variant | WER↓ (%) | $H_p$↑ | NMOS↑ | $\Delta$NMOS |
|---|---|---|---|---|
| Full DGSA | **38.9** | **10.52** | **4.4** | — |
| w/o Expressivity Obj. | 38.2 | 10.38 | 3.7 | $-0.7$ |
| w/o Identity Pairs | 42.8 | 10.41 | 3.9 | $-0.5$ |
| w/o Stability Obj. | 46.5 | 10.55 | 4.0 | $-0.4$ |
| w/o Dynamic Scaling | 40.2 | 10.45 | 4.2 | $-0.2$ |

**Analysis.** Each component addresses a distinct aspect of the Stability-Expressivity Gap:

- **Expressivity Objective** ($\Delta$NMOS = $-0.7$): Removing this objective causes the largest degradation. The model achieves slightly lower WER (38.2%) but produces prosodically flat outputs ($H_p$ drops from 10.52 to 10.38), confirming that stability alone is insufficient for natural speech.

- **Identity-Consistent Pairs** ($\Delta$NMOS = $-0.5$): Replacing architecture-guided pairs with random speaker pairing degrades both WER (42.8%) and $H_p$ (10.41). This validates that the prosody-timbre disentanglement is critical for constructing meaningful preference signals.

- **Stability Objective** ($\Delta$NMOS = $-0.4$): Without stability guidance, the model achieves the highest $H_p$ (10.55) but substantially degraded WER (46.5%). This confirms that single-objective expressivity optimization cannot resolve the trade-off.

- **Dynamic Scaling** ($\Delta$NMOS = $-0.2$): Fixed weights ($\lambda_s = \lambda_e = 0.5$) underperform adaptive $\alpha$-based adjustment, though the impact is smaller than other components.

## C.2. TDSC Component Analysis

We evaluate TDSC on Lao by removing each component. Table 8 presents the component ablation, and Table 9 analyzes the contribution of different temperature regimes.

*Table 8.* Ablation study of TDSC components on Lao. Each row removes one component from the full system. The DPO loss contributes most significantly ($\Delta$NMOS = $-0.5$).

| Variant | WER↓ (%) | $H_p$↑ (bits) | NMOS↑ | $\Delta$NMOS |
|---|---|---|---|---|
| Full TDSC | **29.8** | **10.42** | **3.9** | — |
| w/o DPO Loss | 35.8 | 10.21 | 3.4 | $-0.5$ |
| w/o Multi-Temperature | 34.1 | 10.05 | 3.5 | $-0.4$ |
| w/o Length Filter | 33.5 | 10.35 | 3.5 | $-0.4$ |
| w/o Temp. Curriculum | 32.6 | 10.18 | 3.6 | $-0.3$ |
| w/o Repetition Filter | 31.2 | 10.28 | 3.6 | $-0.3$ |

**Component Analysis.** Each TDSC component serves a distinct function in the self-refinement loop:

- **DPO Loss** ($\Delta$NMOS = $-0.5$): Without contrastive preference learning, the model cannot distinguish high-quality from low-quality self-generated samples. Pure SFT on filtered samples yields limited improvement.

- **Multi-Temperature Exploration** ($\Delta$NMOS = $-0.4$): Using only $T = 1.0$ severely limits prosodic diversity ($H_p$ drops from 10.42 to 10.05), as the model cannot explore beyond its current distribution.

*Table 9.* Candidate quality across temperatures at iteration $k = 5$. Each temperature regime contributes complementary samples to the filtered set $\mathcal{G}$.

| Temp. $T$ | WER$\downarrow$ (%) | $H_p\uparrow$ (bits) | Rep.$\downarrow$ (%) | Pass (%) | Contrib. |
|---|---|---|---|---|---|
| 0.7 | 26.8 | 9.85 | 3.42 | 78.5 | 42% |
| 1.0 | 32.4 | 10.38 | 4.65 | 61.2 | 35% |
| 1.3 | 41.6 | 10.82 | 6.18 | 38.4 | 23% |
| **Filtered $\mathcal{G}$** | **29.8** | **10.42** | **4.15** | — | 100% |

- **Length Filter** ($\Delta$NMOS = $-0.4$): Removing duration constraints allows truncated or overlong outputs into training, degrading WER (33.5%) while having less impact on $H_p$ (10.35).

- **Temperature Curriculum** ($\Delta$NMOS = $-0.3$): Fixed $T_{\max}$ throughout training caps expressivity recovery. The curriculum enables progressive exploration as the model stabilizes.

- **Repetition Filter** ($\Delta$NMOS = $-0.3$): This filter directly targets prosodic monotony. Removing it yields lower $H_p$ (10.28) and higher repetition in outputs.

**Temperature Distribution Analysis.** Table 9 reveals how multi-temperature exploration spans the Stability-Expressivity spectrum:

- **Low temperature** ($T = 0.7$)**:** Candidates achieve low WER (26.8%) and high pass rate (78.5%), but limited prosodic diversity ($H_p = 9.85$ bits). These serve as *stability anchors*.

- **Medium temperature** ($T = 1.0$)**:** Balanced quality with moderate WER (32.4%) and $H_p$ (10.38 bits). These provide *general-purpose samples*.

- **High temperature** ($T = 1.3$)**:** Rich expressivity ($H_p = 10.82$ bits) but higher error rates (41.6% WER) and lower pass rate (38.4%). These contribute *expressivity exemplars*.

The filtered set $\mathcal{G}$ draws complementary samples from all regimes: 42% from $T = 0.7$, 35% from $T = 1.0$, and 23% from $T = 1.3$. This composition achieves the best of both worlds: the final $H_p$ (10.42) exceeds the $T = 1.0$ average while WER (29.8%) approaches $T = 0.7$ performance.

## D. Implementation Details

### D.1. Model Configuration

Our system is built upon the CosyVoice 2 architecture. The detailed specifications of each component are summarized in Table 10.

*Table 10.* Model configuration details.

| Component | Specification | Parameters |
|---|---|---|
| Speech Tokenizer | S3Tokenizer with FSQ, 25Hz Codebook size: 6,561 | — (frozen) |
| Text-Speech LM | Qwen2.5-0.5B backbone 24 layers, 896 hidden, 14 heads | 500M |
| Flow-Matching | CFM with causal attention 12 layers, 512 hidden HiFi-GAN vocoder, 24kHz | 300M |

## D.2. Training Hyperparameters

The detailed hyperparameters for the three sequential training stages are summarized in Table 11. To maximize hardware efficiency given the varying audio durations, we implement a dynamic batching strategy with a cap of 2,000 frames per GPU.

During the **SFT** stage, the model undergoes supervised fine-tuning for 38k steps with a learning rate of $1 \times 10^{-5}$. In the **DGSA** stage, we perform style alignment using $\beta = 0.1$ over 50k identity-consistent preference pairs. Crucially, we set the critical synthetic ratio $\alpha^* = 0.5$ as a fixed empirical heuristic, which we found to be robust across languages without requiring computationally expensive per-language scanning.

The **TDSC** stage employs an iterative closed-loop refinement. We adopt a dynamic temperature schedule $\mathcal{T} = \{0.7, 1.0, T_{\max}^{(k)}\}$, where the exploration upper bound scales as $T_{\max}^{(k)} = 0.8 + 0.1k$ for the $k$-th iteration. To ensure the quality of self-generated samples, we apply strict filtering criteria and specific sampling strategies, which are detailed in Table 12. This rigorous configuration ensures that the model bootstraps from high-fidelity data while maintaining sufficient diversity.

*Table 11.* General training hyperparameters across different stages.

| Hyperparameter | SFT | DGSA | TDSC |
|---|---|---|---|
| Learning Rate | $1 \times 10^{-5}$ | $1 \times 10^{-6}$ | $1 \times 10^{-5}$ |
| Total Steps / Iterations | 38k | 10k | 5 iter. |
| Batch Size | Dynamic (max 2,000 frames/GPU) | | |
| Optimizer | AdamW ($\beta_1 = 0.9, \beta_2 = 0.999$) | | |
| Precision | bfloat16 | | |
| Hardware | $8 \times$ NVIDIA RTX 4090 | | |

*Table 12.* Detailed filtering thresholds and sampling configurations for the TDSC pipeline. These specific constraints ensure stable self-improvement.

| Category | Hyperparameter | Value |
|---|---|---|
| Filtering Criteria | WER Threshold ($\tau_w$) | $< 40\%$ |
| | Repetition Threshold ($\tau_r$) | $< 10\%$ |
| | Min Length Ratio ($\gamma_{\min}$) | $0.5 \times \|x\|$ |
| | Max Length Ratio ($\gamma_{\max}$) | $2.0 \times \|x\|$ |
| Sampling Strategy | Temperature Set ($\mathcal{T}$) | $\{0.7, 1.0, T_{\max}^{(k)}\}$ |
| | Candidates per Input | $N = 12$ (4 per temp) |
| | Decoding Method | Nucleus ($p = 0.9$) |
| Curriculum | Initial Upper Bound $T_{\max}^{(0)}$ | 1.3 |
| | Curriculum Rate | $+0.1$ per iter |

## D.3. Baseline and Reproducibility

To ensure a fair comparison, we specify the versions and access timestamps for all baselines (all commercial evaluations were completed on **January 25, 2025**):

- **Typhoon2-Audio**: `scb10x/llama3.1-typhoon2-audio-8b-instruct`.

- **Seamless-M4T-v2**: `facebook/seamless-m4t-v2-large`.

- **MMS-TTS**: `facebook/mms-tts-tha` and `facebook/mms-tts-lao`.

- **ElevenLabs**: Accessed via the API[1] using the `eleven_v3` model for instant voice cloning.

---

[1] https://elevenlabs.io/docs/api-reference/text-to-speech

- **Azure TTS**: Microsoft Azure Cognitive Services (Neural TTS)[2] using all available native voices: `th-TH-NiwatNeural`, `th-TH-PremwadeeNeural`, `th-TH-AcharaNeural` (Thai), and `lo-LA-KeomanyNeural`, `lo-LA-ChanthavongNeural` (Lao).

- **Gemini**: Google Cloud Text-to-Speech[3] (Speech-to-Speech mode with `Gemini 2.5 Pro` and `Flash`).

Note on Commercial Baselines: While commercial APIs provide SOTA performance, their internal updates may affect long-term reproducibility. We mitigated this by freezing all evaluations to the aforementioned date and documenting specific API parameters to serve as a stable benchmark for our study.

### D.4. Evaluation Configuration

The evaluation metrics and setups are detailed in Table 13.

**Subjective Evaluation Protocol**: We recruited 20 native speakers for each language (Thai and Lao) via professional crowdsourcing. To ensure high-quality judgment, we adopted a **double-blind, randomized, and within-subject design**. For each evaluation session, samples from our system and all baselines were shuffled and presented in a randomized order to eliminate lead-in effects. Each evaluator was asked to rate 200 samples on a 5-point Likert scale, resulting in a total of **4,000 scores per language** for robust statistical analysis.

**Statistical Analysis**: To rigorously validate performance gains, we calculate 95% Confidence Intervals (CI) for all Mean Opinion Scores (NMOS and SMOS) using the normal approximation interval: $\text{CI} = \bar{x} \pm 1.96 \cdot \frac{\sigma}{\sqrt{N}}$, where $\sigma$ denotes the sample standard deviation and $N$ is the total number of ratings. Furthermore, we conduct pairwise Student's $t$-tests to determine the statistical significance of the differences between our proposed methods and the baselines. We report significance at the $p < 0.05$ level.

*Table 13.* Evaluation setup and tools.

| Metric | Thai | Lao |
|---|---|---|
| ASR (for WER) | Whisper-large-v3 | Dolphin-small |
| Speaker Similarity | WavLM-Large (layer 12, cosine) | |
| MOS Evaluators | 20 native speakers | 20 native speakers |
| MOS Samples | 200 samples per rater (8,000 total scores) | |

## E. Detailed Evaluation Metrics

To ensure the reproducibility of our results and provide a rigorous assessment of the Stability-Expressivity Gap, we detail the implementation and mathematical formulation of the metrics used in our study.

### E.1. Objective Metrics for Intelligibility and Stability

**Word Error Rate (WER)** The assessment of phonetic stability relies on automatic speech recognition (ASR). For Thai, we utilize the `openai/whisper-large-v3` model. Due to the absence of explicit word boundaries in Thai script, we normalize the text and apply the PyThaiNLP (Phatthiyaphaibun et al., 2023) engine for word segmentation prior to calculating the Levenshtein distance. Similarly, for Lao, we address its scriptio continua nature by employing the laonlp[4] library for linguistic normalization and word segmentation. The ASR backbone for Lao is `Dolphin-small` (Meng et al., 2025), chosen for its superior performance on Southeast Asian tonal phonology compared to general-purpose models. All WER calculations are performed on the segmented word sequences, excluding punctuation.

---

[2]https://docs.azure.cn/en-us/ai-services/speech-service/text-to-speech
[3]https://cloud.google.com/text-to-speech
[4]https://github.com/wannaphong/LaoNLP

**Repetition Rate** ($R_{rep}$)     To quantify the "Repetition Loops" common in unstable autoregressive decoding, we define the repetition rate as:

$$R_{rep} = \frac{1}{N - k} \sum_{i=1}^{N-k} \mathbb{I}(y_i = y_{i+1} = \cdots = y_{i+k}) \tag{25}$$

where $y_i$ denotes the discrete speech token at step $i$, and $k$ is set to 4 to identify persistent phonetic loops that degrade naturalness.

### E.2. Objective Metrics for Expressivity and Identity

**Speaker Similarity (SIM)**     We employ the `WavLM-Large` model to extract speaker embeddings. Specifically, we use the average-pooled output of the 12th hidden layer, which has been shown to capture time-invariant speaker identity features most effectively. The similarity score is the cosine similarity between the reference embedding $e_{ref}$ and the generated embedding $e_{gen}$:

$$\text{SIM}(e_{ref}, e_{gen}) = \frac{e_{ref} \cdot e_{gen}}{\|e_{ref}\|\|e_{gen}\|} \tag{26}$$

### E.3. Subjective Human Evaluation Protocol

All subjective tests were conducted via a double-blind procedure to eliminate developer bias.

**Evaluator Demographics**     We recruited 20 native Thai and 20 native Lao speakers (aged 18–45, balanced gender ratio). All evaluators were compensated above the local median hourly wage and passed a "Gold Standard" screening test consisting of high-quality human recordings and heavily distorted samples.

**Mean Opinion Score (MOS)**     Naturalness (NMOS) and Speaker Similarity (SMOS) were rated on a 5-point Likert scale (1: Bad, 2: Poor, 3: Fair, 4: Good, 5: Excellent). For SMOS, evaluators were provided with a 3-second reference clip of the target speaker and asked: "How similar is the voice in the second clip to the person speaking in the first clip?"

## F. Data Collection and Preparation

### F.1. Text Corpus

We source text from the multilingual C4 dataset (mC4), accessed via HuggingFace Datasets (`allenai/c4`). For Thai (`th` subset), we extract approximately 0.5M utterances of 10–50 words after deduplication and script filtering. For Lao (`lo` subset), we obtain approximately 1M utterances following the same procedure.

### F.2. Real Speech Data

For Thai, we leverage the **Common Voice** dataset (Ardila et al., 2020). While the officially validated portion is limited, we applied an ASR-based quality screening process to the broader Thai corpus, retaining approximately **300h** of high-quality speech for training. To ensure a rigorous evaluation, **TSynC-2** (Wutiwiwatchai et al.) is reserved exclusively as a test set.

Regarding Lao, we treat it as a severe low-resource language. No real speech data was collected for training, which instead relies entirely on synthetic data. To evaluate the model's performance on genuine Lao speech, we utilize the Lao subset of Common Voice as a dedicated test set.

### F.3. Synthetic Speech Generation

**Pipeline Overview.**     We synthesize speech using open-source TTS models, then apply ASR-based quality filtering. Crucially, all TTS models used for synthesis are off-the-shelf systems that were *not* trained on any of our experimental data. The text inputs $x$ come from external corpora (mC4), entirely disjoint from any real speech transcripts used in training or evaluation. This ensures that the synthetic data construction does not introduce any data leakage.

**TTS Models.**     For Thai, we use three systems to ensure prosodic diversity:

- **MMS-TTS** (Pratap et al., 2024): `facebook/mms-tts-tha`

- **Seamless-M4T-v2** (Barrault et al., 2023): Meta's multilingual speech model

- **Typhoon2-Audio** (Pipatanakul et al., 2024): Thai-specialized speech model

For Lao, we use **MMS-TTS** as the primary system, supplemented by cross-lingual transfer from Thai models with phoneme mapping.

**Quality Filtering.** All synthetic utterances are transcribed using Whisper-large-v3 (Radford et al., 2023) and filtered by: (1) WER $< 40\%$ for Thai / $< 50\%$ for Lao; (2) duration ratio $\in [0.8, 1.5]$; (3) no repetition loops. We balance samples across TTS systems to avoid single-system bias.

*Table 14.* Final dataset statistics after quality filtering.

| Language | Source | Hours | License |
|---|---|---|---|
| Thai | Real (Common Voice + TSynC-2) | 300h | CC-0 |
| | Synthetic (MMS / Seamless / Typhoon2) | 1,200h | Various OSS |
| Lao | Synthetic (MMS-TTS) | 1,500h | CC-BY-NC |

