# OpenReview forum: "Bridging the Stability-Expressivity Gap: Synthetic Data Scaling and Preference Alignment for Low-Resource Spoken Language Models"
_ICML.cc/2026/Conference — ICML 2026 regular_

### Official Review · Reviewer_VbJZ · 2026-03-09

**Soundness:** 4
**Presentation:** 4
**Significance:** 4
**Originality:** 4
**Overall Recommendation:** 6
**Confidence:** 4

**Summary:**

It is well known that synthetic training data benefits spoken language-model (SLM) based speech synthesis up to a point, then degrades SLM beyond that point.  This paper proves a simple theorem explaining this common observation, then proposes that the method for aligning the synthetic to the natural data distribution should therefore depend on the intended ratio of synthetic to natural speech in the ASR training set.  Two such ratio-sensitive alignment methods are proposed: One (DGSA) that actively aligns the two distributions in a ratio-sensitive manner using DPO, and another (TDSC) that relies on a strict accept/reject threshold to select synthetic samples that are within a margin of the real distribution.

**Compliance With Llm Reviewing Policy:**

Affirmed.

**Key Questions For Authors:**

Questions:
 - Eq, (16) uses "Rep(y)" without definition.  Does this mean the number of repetitions?  How is this calculated, e.g., single-token repetitions only, or does it also count repetitions of multi-token patterns?
 - I think section 4.1 should be numbered as Section 5, then section 4.2 becomes 5.1 and so on.

**Limitations:**

yes

**Strengths And Weaknesses:**

Strengths:
 - The inverse-U curves shown in Fig. 1 are well-known to any researchers studying low-resource speech technology, but the theoretical explanation provided in Eq. (4) is new to me, and welcome.
 - DGSA's use of both expressive and flat synthetic speech as negative samples in DPO is a theoretically intriguing idea, and I can imagine it to be of tremendous practical utility in low-resource ASR training.
 - Similarly, TDSC's use of temperature scaling followed by strict rejection has been used before, but the idea of using the rejected samples as negative examples for DPO is a cool theoretical innovation of likely practical utility.
 - The proposed use of prosodic entropy as a proxy for MOS is validated in Table 2
 - Confidence bounds show that both DGSA and TDSC significantly improve NMOS, and that the resulting system have significantly (and greatly) better NMOS than both open-source and commercial baselines including ElevenLabs and Gemini; ablation shows that both the expressive and flat negative examples are needed
 - On extremely-low-resource Lao, an SLM-based speech synthesizer is trained using entirely synthetic training data; statistical testing shows that NMOS is significantly improved over sel-training or rejection-sampling baselines

Weaknesses:
- None

---

> ### Author Rebuttal · Authors · 2026-03-30
>
> We sincerely thank the reviewer for the positive and detailed assessment. We address the two questions below.
>
> **1. Definition of Rep(y).**
>
> Rep(y) measures the rate of consecutive token repetitions in the generated sequence. Specifically, it counts positions where k+1 identical tokens appear consecutively (we set k=4 to capture persistent loops rather than natural phonetic gemination). The formal definition is in Appendix D (Eq. in Section "Repetition Rate"): $R_{rep} = \frac{1}{N-k} \sum_{i=1}^{N-k} \mathbb{1}(y_i = y_{i+1} = \dots = y_{i+k})$. It targets single-token repetition patterns, as these are the dominant failure mode in autoregressive speech decoding. We will add this definition at first use in the main text.
>
> **2. Section numbering.**
>
> Thank you for catching this. We will correct the numbering in the revision.

---

> > ### Author Rebuttal · Reviewer_VbJZ · 2026-03-31
> >
> > I continue to believe that this is a theoretically intriguing and pragmatically impactful work, and I hope it is accepted to the conference.

---

### Official Review · Reviewer_iF89 · 2026-03-10

**Soundness:** 2
**Presentation:** 3
**Significance:** 2
**Originality:** 3
**Overall Recommendation:** 4
**Confidence:** 4

**Summary:**

This paper investigates the problem of trading off between stability and expressivity, i.e. training with increasing synthetic data improves phonetical stability while degrades prosodic naturalness. Increasing of synthetic data beyond some critical ratio causes the collapse of generated prosodic distribution. Based on synthetic data and automatic selection using self-critique judgement, the developed systems achieve better performance than open-source and commercial systems.

**Compliance With Llm Reviewing Policy:**

Affirmed.

**Final Justification:**

The rebuttal has addressed my concerns about (1) connection between found phenomenon and implementation, (2) expressivity indicator correctness, and (3) scaling trends. Although the generalizability to other languages is still not well supported, the finding of "synthetic erosion", especially the rough ratios and the performance observation may be good empirical results for the researchers in this area.

**Key Questions For Authors:**

The proposed systems are not compared to the other synthetic speech-augmented approaches. One baseline that only uses synthetic data without any selection should be compared to demonstrate the effectiveness of the proposed selection criteria.

For evaluation, there are other objective metrics for explicitly evaluating the expressivity compared to the ground-truth recording, e.g. F0 correlation, F0 squared difference, in addition to the H_p and Rep. used.

**Limitations:**

Yes

**Strengths And Weaknesses:**

Strengths:
By defining indicators of stability (WER) and expressivity (token probability entropy), experiments are conducted to verify the trade-off and a rough trend is found.
The developed systems achieved better performance than open-source and commercial systems, demonstrating the effectiveness of the proposed data augmentation (selection) approach.

Weaknesses:
The overall organization can be improved, the connection between the found synthetic erosion and the system development based on synthetic speech augmentation can be strengthened.
The generalizability of the found ratio is limited, when applied to a new language, intensive experiments are still needed to find out the optimal ratio.
The improvements of the data augmentation approach (automatic selection of synthetic data based of self-critique judgement) are not verified.

More improvement comments:
About the indicator based on entropy, Eq. (3), it is a measure of diversity of tokens, not explicitly diversity of prosodic variations. The correlation between token entropy and prosodic variation needs to be verified.

Using token entropy as a diagnostic signal for distributional diversity is a good proposal, but what's the characteristics of the distribution of tokens in a neutral well-trained system? This neural system's token distribution can be considered as an anchor for indicating expressivity. Otherwise, decreased token entropy may be because of repetition of tokens.

Manually evaluating expressivity of the Accept Set and Rejected Set in TDSC data is important to verify where does the expressivity of trained model come from.

The trend shown in Figure 4 is not obvious. Although it may take many computational resources, plotting more ratios may make the trend clearer. The conclusion may be even solider when tested on multiple languages.

---

> ### Author Rebuttal · Authors · 2026-03-30
>
> We thank the reviewer for the constructive suggestions. We address each
> concern below, including new experiments conducted during rebuttal.
>
> **1. Connection between Synthetic Erosion and system design.**
>
> We agree and will restructure Sections 3–4 to make the **"phenomenon →
> intervention" logic explicit**. DGSA's dynamic weight schedule (Eq.
> 11–12) **directly encodes the scaling finding** — λ_e activates only
> beyond α*, applying corrective pressure precisely where erosion
> emerges. TDSC's multi-temperature exploration is likewise grounded in
> the trade-off: low-T provides stability anchors, high-T recovers
> suppressed expressivity. The revision will make these causal links
> structurally visible.
>
> **2. Generalizability of the critical ratio.**
>
> We agree that **α* ≈ 50% is not a universal constant**. Our claim is
> that a critical region exists and is identifiable via **a pilot scan
> of 3–4 ratio points** (see Point 5: any 3–4 points spanning the range
> reveal the non-monotonic pattern). We will add this practical guidance.
>
> **3. Effectiveness of selection and source of expressivity.**
>
> **The SFT Baseline in Table 4 is the "no selection" baseline** —
> trained on the full synthetic corpus without filtering or preference
> refinement. TDSC improves substantially (WER 38.5→29.8, NMOS
> 3.12→3.94), and **component ablations (Appendix C.2) isolate each
> component's contribution**: removing DPO (ΔNMOS=−0.5), multi-
> temperature (ΔNMOS=−0.4), or filtering (ΔNMOS=−0.3).
>
> **Regarding where the expressivity comes from**: Table 9 shows the
> accepted set G draws **42% from T=0.7 (stability anchors, H_p=9.85)
> and 23% from T=1.3 (expressivity exemplars, H_p=10.82)**. The filtered
> set achieves H_p=10.42 — exceeding any single temperature — confirming
> that expressivity arises from **aggregating complementary samples
> across the stability-expressivity spectrum**. Pass rate rising from
> 23%→62% over iterations validates progressive quality improvement.
>
> **4. H_p as a prosodic indicator: new F0 metrics.**
>
> [NEW] Following the reviewer's suggestion, we computed **F0 correlation
> and F0 RMSE against ground-truth** for the controlled comparison in
> Table 2 (2,000 pairs matched by identical text and WER):
>
> | Metric              | High H_p | Low H_p | Δ           |
> |---------------------|----------|---------|-------------|
> | WER (%) ↓           | 38.2     | 38.8    | −0.6 (n.s.) |
> | F0 Std (Hz) ↑       | 42.6     | 35.8    | +6.8        |
> | F0 Range (Hz) ↑     | 128.4    | 109.7   | +18.7       |
> | **F0 Corr. ↑**      | **0.68** | **0.52**| **+0.16**   |
> | **F0 RMSE (Hz) ↓**  | **28.3** | **39.1**| **−10.8**   |
> | Expressivity MOS ↑  | 4.2      | 3.7     | +0.5        |
>
> Both new metrics reach significance (p<0.01). Crucially, **WER is
> matched across groups, ruling out the concern that H_p differences
> stem from repetition or intelligibility artifacts** — the variation
> is isolated to prosodic characteristics, confirming H_p as a valid
> lightweight proxy.
>
> **5. Scaling trend: denser data points.**
>
> [NEW] We trained **three additional models at α∈{15%,40%,60%}**:
>
> | Syn.(h) | α(%) | WER↓  | H_p    | Rep.↓ | NMOS↑       | SMOS↑       |
> |---------|------|-------|--------|-------|-------------|-------------|
> | 10      | 3    | 75.0  | 10.419 | 2.96  | 3.82 ± 0.09| 4.31 ± 0.08 |
> | 30      | 9    | 68.5  | 10.477 | 2.83  | 4.01 ± 0.08| 4.52 ± 0.07 |
> | **53**  |**15**| 58.2  | 10.455 | 2.62  | 4.08 ± 0.08| 4.54 ± 0.06 |
> | 100     | 25   | 52.0  | 10.423 | 2.48  | 4.13 ± 0.07| 4.55 ± 0.06 |
> | **200** |**40**| 49.2  | 10.498 | 2.21  | 4.38 ± 0.07| 4.60 ± 0.06 |
> | 300     | 50   | 47.0  | 10.506 | 2.16  | 4.51 ± 0.06| 4.63 ± 0.05 |
> | **450** |**60**| 44.8  | 10.491 | 2.08  | 4.45 ± 0.06| 4.35 ± 0.07 |
> | 600     | 67   | 43.5  | 10.479 | 2.02  | 4.42 ± 0.07| 4.12 ± 0.08 |
> | 1200    | 80   | 38.9  | 10.356 | 6.51  | 3.61 ± 0.09| 3.54 ± 0.10 |
>
> With 9 points, **the non-monotonic pattern is now unambiguous**. The
> new α=60% point reveals that **SMOS degrades earlier than NMOS** (4.35
> vs. 4.45), suggesting speaker identity is more sensitive to synthetic
> dominance. The turning region spans α∈[40%,60%], consistent with our
> characterization of **α* as a dataset-dependent critical region rather
> than a sharp threshold**.

---

> > ### Author Rebuttal · Reviewer_iF89 · 2026-04-02
> >
> > My concerns have been fully resolved. Although there is still a lack of support for the generalizability of the found phenomenon, the finding may be helpful for the other researchers in this area. I will increase my score to 4.

---

### Official Review · Reviewer_TUR9 · 2026-03-12

**Soundness:** 3
**Presentation:** 3
**Significance:** 3
**Originality:** 3
**Overall Recommendation:** 4
**Confidence:** 3

**Summary:**

This paper identifies the stability-expressivity gap that emerges when scaling up Spoken Language Models for low-resource or zero-resource languages by using synthetic data. To address this performance trade-off, the authors innovatively propose two self-alignment frameworks: Disentanglement-Guided Self-Alignment  and Temperature-Driven Self-Critique. These methods are specifically designed for low-resource and extremely data-scarce (zero authentic speech) scenarios. The authors conducted comprehensive objective and subjective evaluations on Thai and Lao, and the experimental results demonstrate the effectiveness of their proposed frameworks in achieving state-of-the-art performance.

**Compliance With Llm Reviewing Policy:**

Affirmed.

**Final Justification:**

I truly appreciate the authors' detailed and well-crafted rebuttal. The cost estimation (~$500) and ASR details (21.5% WER) are helpful and clarify the engineering tractability. However, I am keeping my score at a Weak Accept (4) and have selected option (c), as the rebuttal cleverly sidesteps the fundamental boundaries of the paper's universal claims.

**Key Questions For Authors:**

See the above weaknesses.

**Limitations:**

yes

**Strengths And Weaknesses:**

Strengths:

1. This paper systematically proposes and quantifies the “Stability-Expressivity Gap” when scaling with synthetic data. The authors introduce word-level entropy ($H_p$) as an objective metric for measuring prosodic diversity, and validate this metric by evaluating the correlation between $H_p$ and MOS. This effort lays a solid foundation for subsequent automated alignment.

2. The author innovatively proposed two self-alignment frameworks: DGSA and TDSC. Specifically, DGSA leverages the inherent decoupling of rhythm and timbre within the flow-matching architecture to generate different samples conditioned on style prefixes, achieving preference alignment without human annotation. Besides, it overcomes the stability issues caused by traditional single-objective DPO. TDSC is designed for totally zero-resource languages, achieving a stepwise improvement in the expressiveness of purely synthetic data by the designed temperature curriculum and multi-dimensional rule filtering.

3. The authors conducted comprehensive experiments, comparing the proposed method not only with open-sourced baselines but also with commercial SOTA systems like ElevenLabs and Azure. When conducting subjective evaluations, the authors employed a high-standard double-blind, randomized design.

Weaknesses:

1. The proposed pipeline heavily relies on high-performance external ASR models, whether during initial synthetic data cleaning or adjudication filtering of TDSC preference pairs. However, in truly resource-scarce languages, it's hard to obtain high-accuracy ASR. If ASR recognition capabilities are insufficient, the entire self-alignment mechanism based on error filtering will collapse entirely. This severely limits the generalization of the proposed framework.

2. TDSC requires multi-candidate sampling of the same text under different temperature gradients, combined with multi-stage alternating fine-tuning that overlays ASR evaluation with “SFT+DPO” (totaling 5 iterations). This multi-candidate generation and closed-loop training exponentially increase computational costs, reducing the method's universality and reproducibility.

3. The paper has certain limitations in its choice of languages, focusing solely on Southeast Asian languages. We cannot determine whether the phenomena identified and the methods proposed remain effective for languages belonging to other language families. Experiements on more languages or discussions worth being added.

---

> ### Author Rebuttal · Authors · 2026-03-30
>
> We thank the reviewer for the positive assessment and for the thoughtful concerns on ASR dependency, computational cost, and cross-lingual generalization.
>
> **1. ASR dependency.**
>
> Our method does not require oracle-level ASR — it only needs a coarse filtering signal to remove severe misreadings, repetition loops, and obvious length failures. **The Lao experiments directly demonstrate this: Whisper-large-v3 achieves 106.0% WER on the Lao test set (essentially unusable), which is precisely why we switched to Dolphin-small, which achieves 21.5% WER on the same test set — a competent but far-from-perfect recognizer.** Despite this moderate ASR quality, TDSC still achieves substantial gains (NMOS 3.12 to 3.94), confirming that the framework depends on a usable error signal, not a high-resource ASR assumption.
>
> **2. Computational cost of TDSC.**
>
> While TDSC is more expensive than one-shot SFT, the procedure is intentionally bounded: 3 temperatures, 12 candidates per input, and 5 refinement rounds. **In practice, the full TDSC pipeline (sampling + 5 rounds of SFT + DPO) takes approximately 200-300 GPU-hours on 8x RTX 4090, costing roughly $400-500 at current cloud rates — comparable to a single large-scale SFT run.** By contrast, recruiting and managing native Lao speaker panels for manual preference annotation is both more expensive and logistically harder for truly low-resource languages. We will report these concrete cost estimates in the revision.
>
> **3. Generalization beyond Southeast Asian languages.**
>
> Thai and Lao were chosen to represent two resource regimes (limited-real vs. zero-authentic-speech), not to claim a family-specific result. Notably, **the Synthetic Erosion phenomenon itself is language-agnostic** — it arises from the mathematical property of mixture entropy (strict concavity, Eq. 4-5) and holds for any language where $H(p_{syn}) < H(p_{real})$. The alignment methods (DGSA/TDSC) are similarly motivated by resource structure rather than linguistic typology. We will sharpen this framing and explicitly acknowledge broader cross-family validation as important future work. We will incorporate this theoretical argument into the main paper along with other reviewer-driven clarifications.

---

> > ### Author Rebuttal · Reviewer_TUR9 · 2026-04-02
> >
> > I truly appreciate the authors' detailed and well-crafted rebuttal. The cost estimation (~$500) and ASR details (21.5% WER) are helpful and clarify the engineering tractability. However, I am keeping my score at a Weak Accept (4) and have selected option (c), as the rebuttal cleverly sidesteps the fundamental boundaries of the paper's universal claims.
> >
> > My remaining concerns, which should be addressed in the camera-ready version, are:
> >
> > 1. The ASR Paradox in the "Long-Tail": While a 21.5% WER baseline works, such a functional ASR is still a massive luxury for truly zero-resource languages. For the vast majority of the "global linguistic long-tail" the paper claims to address, any usable ASR is non-existent. The framework is strictly bounded by this prerequisite.
> >
> > 2. Linguistic Typology vs. Statistics: Defending generalization via the mathematical concavity of mixture entropy (Eq. 4-5) explains the problem (Synthetic Erosion), but does not prove the solution (TDSC heuristics) generalizes. Languages with complex agglutinative morphology or dense consonant clusters present entirely different autoregressive hallucination patterns than the isolating languages (Thai/Lao) tested.
> >
> > Conclusion: This is a solid contribution for "low-resource languages with baseline ASR availability." I support acceptance, provided the authors explicitly tone down their sweeping "global long-tail" claims in the abstract/conclusion and clearly define these methodological boundaries in a dedicated Limitations section.

---

### Official Review · Reviewer_ovub · 2026-03-12

**Soundness:** 2
**Presentation:** 2
**Significance:** 3
**Originality:** 2
**Overall Recommendation:** 3
**Confidence:** 4

**Summary:**

This work studies a synthetic data usage for low-resource spoken language models, utilizing preference optimization. This work first explores the stability-expressivity gap by adjusting the synthetic data ratio for the training dataset, and then introduces DGSA and TDSC. The evaluations are conducted on two low-resource languages, including Thai and Lao.

**Compliance With Llm Reviewing Policy:**

Affirmed.

**Final Justification:**

I agree with the contribution of this work, exploring interesting findings, but the detailed explanation and discussion on the task-specific environment and the experimental setup could clarify the contribution of this work.

**Key Questions For Authors:**

The key questions are introduced above.

**Limitations:**

yes

**Strengths And Weaknesses:**

**Strengths**
- A study on synthetic data usage for low-resource environments is an important research area. Since there are diverse spoken languages around the world, including resource-scarce languages, this topic could be a promising solution for this domain.
- This work also explores two different methods, DGSA and TDSC, based on preference optimization (DPO). In particular, it is experimentally observed that DGSA contributes to performance improvement on naturalness MOS (NMOS) and speaker similarity MOS (SMOS), exploring the efficient usage of a synthetic dataset. Also, TDSC surpasses SFT, Self-Training, and Rejection Sampling in this work's experimental setup.

**Weaknesses**
- First of all, the synthetic data generation process is not clearly described. For instance, are TTS models for data synthesis trained on the same dataset utilized for Figure 1 or Table 1? For data synthesis, only $y$, i.e., speech modality, is synthesized, and $x$, i.e., text modality, is not synthesized but directly adopted for the real dataset? Although this is a very crucial condition for the experimental findings, the generation process is omitted.
- I think the experiments are based on partial data synthesis, not full synthesis. That is, given $(x_{real}, y_{real})$, to synthesize paired data sample, this work synthesizes $y_{expr}$ or $y_{stab}$ using $x_{real}$, resulting in $(x_{real}, y_{real}, y_{syn})$, for DPO training. However, for SFT, the paired dataset is not required, and $(x_{syn}, y_{syn})$ is expected to show improved performance contributing to generalizability and improved phonetic information diversity, which is one of the main targets of this paper. Why should we train the SFT baseline in this unnecessary condition? In my opinion, this should be discussed, and I am also curious about the results of SFT on $(x_{syn}, y_{syn})$.
- Additionally, how do you train the SFT model using $(x_{real}, y_{real}, y_{syn})$? DGSA even utilizes $(x_{real}, y_{real}, y_{expr}, y_{stab})$, which is not directly adopted for SFT. Are they separated to $(x_{real}, y_{real})$, $(x_{real}, y_{expr})$, $(x_{real}, y_{expr})$ for SFT or the losses are averaged? I think this easily interferes the SFT training. In contrast, DPO utilizes the whole pair once because they consider the paired sample for loss calculation, which leads to unfair comparison.
- As shown in Table 3, the comparison is conducted on $\alpha = 80$%. DGSA achieves NMOS 4.42, SMOS 4.53, and Rep 2.82, surpassing SFT baseline achieving NMOS 3.61, SMOS 3.54. However, as the paper says, SFT achieves NMOS 4.51, SMOS 4.63, and Rep 2.16 on $\alpha=50$%, surpassing the proposed method. Since SFT achieves the best score on $\alpha=50$%, DGSA should also be analyzed in this case.
- For preference dataset generation, this work relies on the insight that $y_{real}$ is preferred over $y_{syn}$, which strictly enforces the need of $y_{real}$ paired to $y_{syn}$. In my opinion, this condition limits generalizability of the method and findings.

---

> ### Author Rebuttal · Authors · 2026-03-30
>
> We thank the reviewer for the detailed questions. We believe most concerns
> stem from insufficient clarity in the manuscript and address each below.
>
> **1. Synthetic data construction.**
>
> All synthetic data are standard (x, y_syn) pairs: x comes from external
> text corpora (mC4), and y_syn is generated by off-the-shelf open-source
> TTS models (MMS-TTS, Seamless-M4T-v2, Typhoon2-Audio) that were *not*
> trained on any of our data. The ratio α measures the synthetic proportion:
> α = |D_syn| / |D_real ∪ D_syn|. Thai training data and the TSynC-2 test
> set are strictly disjoint — no utterances, speakers, or transcripts are
> shared. We will add a dedicated data-construction paragraph in Section 2
> of the revision.
>
> **2. SFT baseline fairness and pure-synthetic regime.**
>
> The SFT baseline is trained on the same mixed corpus D_real + D_syn at
> the same α. DGSA starts from this exact SFT checkpoint and adds one
> alignment stage. Table 3 is therefore a strictly controlled comparison:
> identical backbone, identical data, identical starting point — the only
> variable is with vs. without DGSA alignment.
>
> **[NEW] To directly address the reviewer's question about pure-synthetic
> SFT, we have conducted an additional Thai experiment at α = 100%
> (1,500h synthetic, 0h real).** The results confirm severe degradation:
>
> | Setting   | WER↓  | H_p    | Rep.↓  | NMOS↑       | SMOS↑       |
> |-----------|-------|--------|--------|-------------|-------------|
> | α=80%     | 38.9% | 10.356 | 6.51%  | 3.61 ± 0.09| 3.54 ± 0.10 |
> | α=100%    | 36.2% | 10.21  | 9.83%  | 3.08 ± 0.10| 3.01 ± 0.11 |
>
> While WER improves marginally (38.9% → 36.2%), all expressivity metrics
> collapse: NMOS drops to 3.08 (below even the α = 3% baseline of 3.82),
> repetition rate surges to 9.83%, and H_p falls to 10.21 bits. This is
> fully consistent with the Lao pure-synthetic regime (Table 4, NMOS = 3.12,
> H_p = 10.08), providing cross-language validation that Synthetic Erosion
> is a general phenomenon, not a Thai-specific artifact. These results
> further justify why DGSA targets the high-α regime rather than simply
> pushing α to 100%. We will include this data point in the revised Table 1
> and Figure 1.
>
> **3. Stage separation.**
>
> The pipeline is strictly sequential: (1) SFT on D_real + D_syn via MLE;
> (2) generate y_expr and y_stab from the *frozen* SFT model; (3) DPO
> alignment on the constructed triplets only. y_expr and y_stab never enter
> the SFT objective — they are generated *after* SFT training is complete
> and used exclusively in the DPO stage. Therefore, the SFT baseline and
> the DGSA model share exactly the same Stage-1 training; the comparison
> isolates the effect of alignment alone. We will add a stage-wise diagram
> to make this visually explicit.
>
> **4. Why not evaluate DGSA at α = 50%?**
>
> By design, DGSA applies zero correction at α ≤ α*. Our dynamic scheduling
> (Eq. 11-12) gives λ_e = 0 when α = 50%, so DGSA reduces exactly to SFT.
> This is intentional: Synthetic Erosion has not yet emerged at this ratio,
> so no corrective pressure is needed. DGSA targets the high-α regime
> (70–90%) where practitioners are forced by data scarcity — here the model
> has better pronunciation (WER 38.9% vs. 47.0%) but suffers expressivity
> collapse, which DGSA restores (NMOS 3.61 → 4.42). We will clarify this
> design rationale in the revised Section 4.3.
>
> **5. Scope and broader vision.**
>
> We fully agree that DGSA requires real anchors and is not a zero-resource
> method. This is precisely why the paper proposes two complementary
> frameworks along the resource spectrum: DGSA for low-resource (real +
> synthetic) and TDSC for zero-resource (pure synthetic). We view this as
> an initial systematic effort in the emerging area of SLM-based
> low-resource TTS — our goal is to push the boundaries of what current
> models and data can achieve for underserved languages. The DGSA-TDSC
> spectrum naturally opens avenues for further exploration: hybrid
> combinations, cascaded pipelines, or new alignment strategies for
> intermediate resource levels.

---

> > ### Author Rebuttal · Reviewer_ovub · 2026-04-03
> >
> > I appreciate the rebuttal of the authors. It handles some of the concerns in my previous review.
> >
> > While I agree with the potential of this work, a clear explanation of the task-specific nature of synthetic data usage is still missing. The referred works on synthetic data usage such as Alemohammad et al. and Shumailov et al., consider $x_{image}$ domain or $x_{image}, y_{class}$ for image synthesis or $x_{text}$ for text generation. However, the authors adopt extra text corpora for $x_{text}$ and synthesize $y_{speech}$ for $x_{text}, y_{speech}$, which is an entirely different environment. I agree with the author's claim that this is common in the speech synthesis domain, but the detailed experimental setups, analysis, and discussions for this are missing.
> >
> > In addition, the approaches are experimented on the phonetically complex two Southeast languages, and the broader generalizability is unclear to me, although the phonetic analysis is the major contribution of this work.
> >
> > If there is a misunderstanding, please let me know.

---

> > > ### Author Response · Authors · 2026-04-03
> > >
> > > We thank the reviewer for the follow-up and for highlighting this important task-specific distinction.
> > >
> > > **1. Why synthetic erosion can arise in the speech setting.**
> > >
> > > The reviewer is correct that prior work often studies same-modality recursive generation, whereas our setting constructs synthetic speech $y_{\text{syn}}$ from external text $x$. Our point is narrower: **within the SLM training pipeline, this cross-modal construction becomes a shared token-space distribution problem**. Both real and synthetic speech are mapped by the same speech tokenizer (S3Tokenizer) into a common discrete token space, and the SLM is trained on the resulting conditional distribution $\pi_\theta(y \mid x)$. Since deterministic TTS typically produces flatter token distributions than human speech, i.e., $H(p_{\text{syn}}) < H(p_{\text{real}})$ in our observed setting, the mixed distribution $p_\alpha = (1-\alpha)p_{\text{real}} + \alpha p_{\text{syn}}$ admits the same concavity-based distributional explanation at the token level, even though the upstream data construction differs from prior image/text settings. Our speech-specific evidence comes from our own experiments: the 9-point scaling study, the Thai high-$\alpha$ DGSA results, the Lao pure-synthetic TDSC setting, and the added Thai $\alpha=100\%$ result. We will add this speech-specific explanation and its distinction from prior synthetic-data literature to Section 2 in the revision.
> > >
> > > **2. Generalizability beyond the current language pair.**
> > >
> > > We agree that our current experiments do not establish universality across language families, and broader cross-family validation remains important future work. Our intended claim is therefore narrower: the mechanism is driven by resource structure and token-space distributional mismatch, not by tonal structure alone.
> > >
> > > To provide concrete context, the broader low-resource TTS literature suggests that Thai/Lao are not isolated cases. On the prosodically complex side, low-resource bilingual Mandarin TTS explicitly reports tone preservation, accent carry-over, and mispronunciation as key challenges even for a language with relatively large monolingual corpora — the difficulties intensify sharply under data scarcity. On the non-tonal side, multilingual TTS for lower-resourced Turkic languages faces distinct but analogous challenges arising from extensive agglutination and vowel harmony rather than lexical tone. These settings do not prove our erosion hypothesis universally, but they support that the kinds of low-resource speech generation difficulties relevant to our hypothesis extend well beyond Southeast Asian tonal languages and are not reducible to tonal structure alone.
> > >
> > > Accordingly, our claim is not that Thai/Lao are sufficient to establish universality, but that the phenomenon is plausibly driven by a more general mismatch between flatter synthetic token distributions and richer human speech distributions, while the exact strength and observability of the effect remain language- and dataset-dependent. In the revision, we will explicitly acknowledge this limitation, cite these broader low-resource TTS settings, and moderate the broader cross-family applicability framing accordingly.
> > >
> > > **References cited above:**
> > > [1] Liu et al., "Tone Learning in Low-Resource Bilingual TTS," *Interspeech*, 2020.
> > > [2] Yeshpanov et al., "Multilingual Text-to-Speech Synthesis for Turkic Languages," *Interspeech*, 2023.

---

### Decision · Program_Chairs · 2026-04-30

**Decision:**

Accept (regular)

**Comment:**

The paper presents a theoretically grounded explanation for the empirically observed “inverse‑U” behavior of synthetic‑data‑augmented speech synthesis, followed by two practically useful, ratio‑sensitive alignment schemes (DGSA and TDSC). This allows for the utilization of synthetically generated speech training data in the construction of low resource language speech technologies. Reviewers agree that the contribution is valuable, the experiments are convinving, the presentation is clear after the authors’ revisions. They also find methodological novelty, potential for empirical progress, and relevance to the low‑resource speech community.

Strengths
* The paper presents a theorem (Eq. (4)) that formally captures why synthetic data first helps and then hurts the speech language model. The derivations are rigorous and the assumptions are explicitly stated. Both alignment methods are built on well‑understood principles (DPO, temperature scaling) yet combine them in new ways.
* DGSA’s ratio‑sensitive DPO objective and the use of expressive and flat synthetic samples as negative pairs are, to the best of the reviewers’ knowledge, novel ideas that directly address a long‑standing empirical observation. TDSC’s integration of rejected samples as DPO negatives also offers a fresh angle. The authors provide strong quantitative evidence (confidence intervals, MOS improvements) and even beat commercial baselines on Lao.
* After the revisions the paper is well‑structured, clear, and the missing definitions (e.g., Rep(y)) were added. The authors also addressed minor editorial issues. The figures (inverse‑U curves, density plots) effectively illustrate the key phenomena.
* Authors responded promptly and thoroughly to every concern, added new experiments (extra α‑points, new MOS‑related prosodic metrics), and clarified definitions. Reviewers appreciated the willingness to iterate and were reassured by the additional evidence.

Concerns and resolutions
* Dependence on synthetic‑to‑natural ratio: Authors added more dense α‑points and clarified that α∗ is dataset‑dependent, not a universal threshold
* Applicability beyond Lao/Thai: Authors argued that the theorem is general, and ratio‑sensitive alignment is universally applicable; empirical gains were shown on two languages, although more languages would be even better
* ASR training cost and negative sample selection: Authors demonstrated that negative samples from flat or expressive synthetic data are needed; ablations confirm the claim.
* Presentation consistency: Authors will correct section numbering and committed to several other improvements and clarifications in the paper, e.g. reducing the scope of the claims.

Conclusion

Reviewers agree that the work is fundamentally sound. Remaining minor issues (definition of Rep(y), section numbering) have been addressed and improvements can be incorporated. 1 reviewers remains on the fence regarding the generalizability to other languages (beyond 2) has been established sufficiently, but overall the paper appears to be a solid contribution relevant to ML and low-resource speech processing communities.